# Hierarchical motor adaptations negotiate failures during force field learning

Tsuyoshi Ikegami[1,2,3☯]*, Gowrishankar Ganesh[1,2,4☯], Tricia L. Gibo[2,5], Toshinori Yoshioka[2], Rieko Osu[2,6], Mitsuo Kawato[2]

**1** Center for Information and Neural Networks (CiNet), National Institute of Information and Communications Technology (NICT), Osaka, Japan, **2** Brain Information Communication Research Laboratory Group, ATR, Kyoto, Japan, **3** Graduate School of Frontier Biosciences, Osaka University, Osaka, Japan, **4** Centre National de la Recherche Scientifique (CNRS), Universite Montpellier (UM) Laboratoire d'Informatique, de Robotique et de Microelectronique de, Montpellier (LIRMM), Montpellier, France, **5** Emergo by UL, Utrecht, The Netherlands, **6** Faculty of Human Sciences, Waseda University, Saitama, Japan

☯ These authors contributed equally to this work.
* ikegami244@gmail.com

**Data Availability Statement:** Data and codes for all experiments and simulations are freely available in the Dryad repository at the URL: https://doi.org/10.5061/dryad.5x69p8d2f [75].

## Abstract

Humans have the amazing ability to learn the dynamics of the body and environment to develop motor skills. Traditional motor studies using arm reaching paradigms have viewed this ability as the process of 'internal model adaptation'. However, the behaviors have not been fully explored in the case when reaches fail to attain the intended target. Here we examined human reaching under two force fields types; one that induces failures (i.e., target errors), and the other that does not. Our results show the presence of a distinct failure-driven adaptation process that enables quick task success after failures, and before completion of internal model adaptation, but that can result in persistent changes to the undisturbed trajectory. These behaviors can be explained by considering a hierarchical interaction between internal model adaptation and the failure-driven adaptation of reach direction. Our findings suggest that movement failure is negotiated using hierarchical motor adaptations by humans.

## Author summary

How do we improve actions after a movement failure? Although negotiating movement failures is obviously crucial, previous motor-control studies have predominantly examined human movement adaptations in the absence of failures, and it remains unclear how failures affect subsequent movement adaptations. Here we examined this issue by developing a novel force field adaptation task where the hand movement during an arm reaching is perturbed by novel forces that induce a large target error, that is a failure. Our experimental observation and computational modeling show that, in addition to the popular 'internal model learning' process of motor adaptations, humans also utilize a 'failure-negotiating' process, that enables them to quickly improve movements in the presence of failure, even at the expense of increased arm trajectory deflections, which are subsequently reduced gradually with training after the achievement of the task success. Our results

**Funding:** TI was supported by JSPS KAKENHI Grant #26750387 (https://www.jsps.go.jp/j-grantsinaid/). RO was supported by KAKENHI Grants #17H02128 and # 20H05482. MK was supported by AMED Grant #JP20dm0307008 (https://www.amed.go.jp/), and JST ERATO Grant # JPMJER1801 (https://www.jst.go.jp/). TY was supported by a contract with the National Institute of Information and Communications Technology, entitled 'Development of network dynamics modeling methods for human brain data simulation systems' (https://www.nict.go.jp/). The funders had no role in study design, data collection and analysis, decision to publish, or preparation of the manuscript.

**Competing interests:** The authors have declared that no competing interests exist.

suggest that a hierarchical interaction between these two processes is a key for humans to negotiate movement failures.

## Introduction

Imagine you are practicing golf shots in a driving range and aiming to land the ball on the green with a pre-planned ball trajectory. When the ball goes along a different, unintended trajectory but it still lands on the green, you will almost automatically correct your next hitting action, by accounting for the error in the ball trajectory. However, the correction you make will be very different if the ball goes out of bounds of the green. In which case, you would not just make a large correction in the hitting action but also maybe even change your plan of the trajectory. Going out of bounds is considered a failure in golf, penalized by an extra shot, and the movement adaptation by humans in the presence of failure is intuitively very different from when a movement has achieved its target.

Failure-driven adaptations by humans have been extensively studied in decision making or cognitive control [1,2], while it has remained unclear how such distinct adaptations driven by failure affect human motor adaptation. Previous studies on motor adaptation have mainly focused on the internal model adaptation that is driven by sensory prediction error (SPE)–the difference between sensory feedback and sensory prediction of a movement [3–5], and/or motor command error [6]. In the last decade, however, there is mounting evidence that failure or target error (TE)–the difference between the sensory feedback of the movement endpoint and the target position–has a distinct, important contribution to motor adaptation [7,8]. The most popular TE-driven (or failure-driven) motor adaptation process is *explicit strategy learning* [7,9,10], which has been mostly examined during arm reaching adaptation to visuomotor rotations and often quantified by explicit reports of the reaching aiming point [9]. The explicit strategy learning is thought to modify motor performance to reduce TE, independently of SPE [7].

It however remains unclear what is the relation between the TE-driven motor adaptation and the SPE-driven motor adaptation (i.e., internal model adaptation). The interaction between the explicit strategy learning and the internal model adaptation is popularly explained by a two-state model of sensorimotor learning with different time scales for each state [11], where the two operate in a 'flat' (non-hierarchical) manner and the net adaptation is defined to be the sum of the two [9,12]. The fast component of the model has been often suggested to be linked to explicit strategy learning in visuomotor rotation tasks [9,10] as well as force field tasks [10,13,14]. On the other hand, recent studies have shown that the TE modulates the adaptation rate of the SPE-driven internal model adaptation [15] or savings [16]. This role of the TE as a modulator to the internal model adaptation may suggest a hierarchical interaction between the TE-driven and the SPE-driven motor adaptations.

Here we show that the two adaptation processes, in fact, interact hierarchically using a force adaptation paradigm with new *TE-inducing* force fields that perturbed the participant's hand with large forces near the target (Fig 1B). The development of these new fields was crucial, as the force fields used in most reaching adaptation studies induce minimal TE or failure. For example, the popular velocity-dependent curl force-field (VDCF) [17,18] exerts the largest force perturbation on hand movements of participants in the middle of the reach and minimal perturbation near the target during reaching movements with a bell-shaped velocity profile [19]. The force field, thus, results in large lateral deviations (LDs) mid-reach in the early

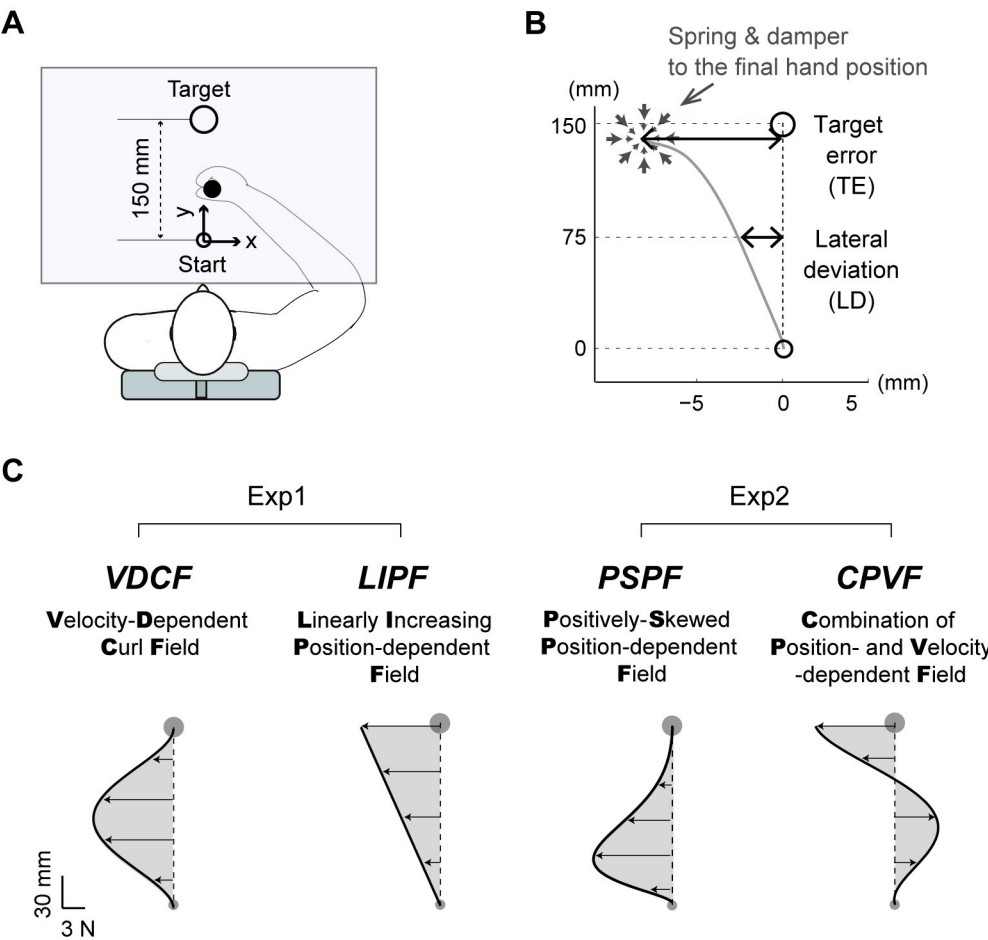

**Fig 1.** Experiment and force fields: A) Participants made a reaching movement from a start point to a target point while holding a handle of a robot manipulandum. The direct vision of the participant's hand was occluded by a table while they received visual feedback of their hand position during each trial by a cursor projected on the table. B) A very stiff two-dimensional spring, which was activated when the hand velocity decreased below a threshold of 20 mm/s, ensured that the participant could not make a second corrective movement to reach the target. C) The reaching task was performed in two force fields in Experiment-1 (VDCF and LIPF) and two force fields in Experiment-2 (PSPF and CPVF). The hand force profiles in these force fields are shown as shaded regions while assuming a straight minimum-jerk hand trajectory along x = 0. VDCF is a velocity-dependent force field, while LIPF and PSPF are position-dependent force fields. CPVF is a linear combination of VDCF and LIPF. Please refer to the methods for the mathematical definitions of the fields.

adaptation trials, but allows the participant to reach their target even after this large LD (see Fig 2A).

In our study with the novel TE-inducing force fields, we observed that TE-driven motor adaptation occurs faster than internal model adaptation. Second, and importantly, TE-driven motor adaptation can result in persistent after-effects that are distinct from after-effects after internal model adaptation. Third, these adaptive behaviors can be well explained by previous models of internal model adaptation only if they incorporate a hierarchical interaction between TE-driven adaptation of the kinematic plan and internal model adaptation. The relation between TE-driven adaptation and internal model adaptation is consistent with the traditional view of hierarchical motor planning of kinematics and dynamics [6,20].

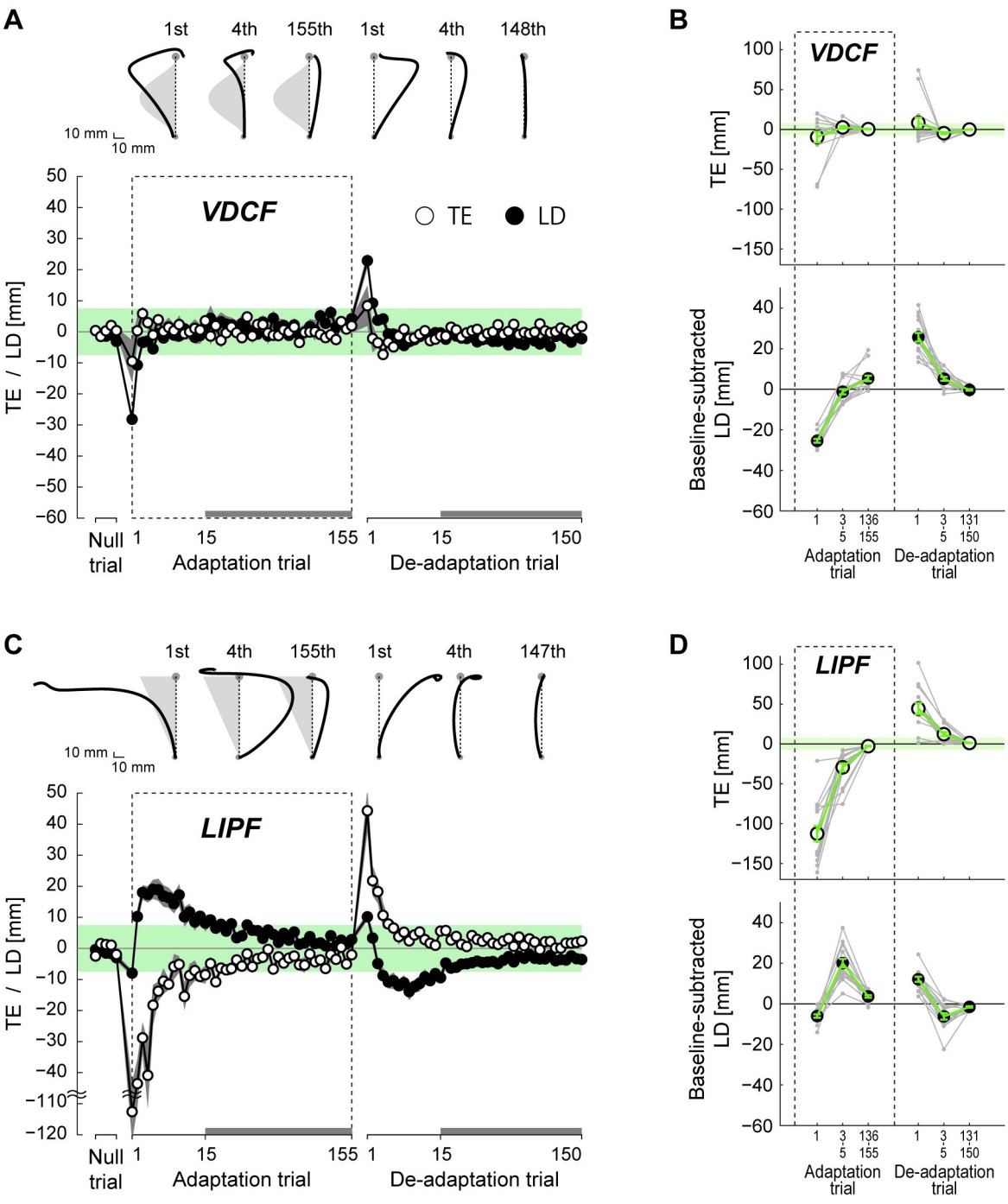

**Fig 2. Trajectory adaptation in Experiment-1:** (A, C) The hand trajectories of two representative participants and learning curves in VDCF (A) and LIPF (C) averaged across all participants. Note that the scales differ between x and y axes to clearly show trajectory changes along the x-axis. The light gray shades behind some trajectories represent a schematic image of the force field. The adaptation of the TE and LD are shown by traces with open circles and filled circles, respectively. The first 15 TE and LD values are plotted for every single trial, while the subsequent trials (indicated by thick gray lines at the bottom of the figure) are plotted for every five trials. The shaded gray areas around the lines represent standard errors. The light green zones represent the target width (radius: 7.5 mm). (B, D) The TEs and baseline-subtracted LDs in six trial epochs (1st, 3rd-5th, 136th-155th adaptation trials, and 1st, 3rd-5th, 131st-150th de-adaptation trials) in VDCF (B) and LIPF (D). Gray dots represent data from individual participants. The error bars indicate standard errors. The light green zone in the TE plots represents the target width.

## Results

### Experiment-1

In Experiment-1, thirty participants were asked to make arm reaching movements to a target 150 mm from the start position (Fig 1A) and adapt to either of two force fields (Fig 1C): the popular velocity-dependent curl field (VDCF) that does not induce TEs, and the novel and TE-inducing linearly increasing position-dependent (orthogonal) field (LIPF) (see Methods for details). The adaptation phase (155 trials) was followed by the de-adaptation phase (150 trials), where the participants performed the same task in the null field, like the baseline session. We randomly assigned the participants to one of the two force fields (n = 15 for each). Their movements were quantified by two variables: TE and LD. The TE was defined as the x-deviation of the endpoint hand position from the target, and the LD was defined as the x-deviations of the hand from the mid-point (y = 75 mm) of the straight line connecting the start and the target (Fig 1A and 1B, and see Methods).

**TE changes the trajectory adaptation pattern.** Fig 2 shows the time development of hand trajectories, TE (open circle), and LD (filled circle) in the two force fields and subsequent null field. To show immediate and later effects of the initial TE on the adaptation and de-adaptation phases, we analyzed the data in six trial epochs: $1^{st}$, $3^{rd}$-$5^{th}$, $136^{th}$-$155^{th}$ (i.e., last 20) adaptation trials and the $1^{st}$, $3^{rd}$-$5^{th}$, $131^{st}$-$150^{th}$ (i.e., last 20) de-adaptation trials (Fig 2B and 2D).

In the VDCF, the trajectory adaptation pattern was similar to those reported in previous studies. The force field perturbed the participants' hand trajectories considerably in the first adaptation trial (Fig 2A), but their hands still could reach the target as we expected. After the adaptation phase, the participants could fully compensate for the perturbation, and their trajectories became straighter, curving towards the opposite direction by the $155^{th}$ adaptation trial. In the first de-adaptation trial, their hand trajectories exhibited a large *after-effect*, deviating towards the opposite direction to the force field. By the end of the de-adaptation phase, their trajectories returned to the straight baseline, or *null*, trajectories (see $148^{th}$ de-adaptation trial). These results were consistent with what has been observed in previous studies [18,21]. The across-participant average adaptation of the TE (open circle) and LD (filled circle) are shown in the bottom panels of Fig 2A. A large LD induced at the beginning of the adaptation and de-adaptation phases quickly decreased to within the target size (radius = 7.5 mm, light green zone) within the first 10 adaptation and de-adaptation trials, respectively. Importantly, TEs remained relatively small—around or within the target from the very first adaptation trial and through the following adaptation and de-adaptation phases. In fact, the magnitude of TE was not significantly larger than the target radius in the first adaptation trial (t(14) = 0.284, p = 0.780) and the first de-adaptation trials (t(14) = 0.131, p = 0.897).

On the other hand, the TE-inducing LIPF showed a dramatically different adaptation pattern from the VDCF. In the LIPF, the participants' hand trajectories in the first adaptation trial (Fig 2 C) were perturbed the most around the target, resulting in a large TE (across-participants average of TE in $1^{st}$ trial was 112.6 ± 38.0 (mean ± s.d.) mm) that was significantly larger than the target (t(14) = 10.700, p = 4.016×$10^{-8}$). In the subsequent adaptation trials (see $4^{th}$ adaptation trial in Fig 2C), the participant's hand trajectories jumped opposite to the force direction, which ensures that the target is reached, even with a curved trajectory. It is important to note that the magnitude of the LD increases (between $2^{nd}$ and $7^{th}$ adaptation trials), before it gradually decays after the $7^{th}$ adaptation trial. Furthermore, the decay was observed to be opposite in sign to that in VDCF. That is, while the LD in the VDCF decays from an initial negative value (i.e., from '–x' towards zero), the decay in the LIPF is from a positive deviation (i.e., from '+x' towards zero), even though the LIPF also pushes the hand in the same direction

as the VDCF field (i.e., towards '–x'). Consequently, the decays of the TE and LD are of the same sign in the VDCF, but opposite signs in the LIPF.

The trajectory change in the de-adaptation phase ($1^{st}$, $4^{th}$, and $147^{th}$ de-adaptation trials in Fig 2. C) was almost a mirror image of that in the adaptation phase. A distinctly large TE (of 44.3 ± 27.7 mm) was induced in the first de-adaptation trial, which was again significantly larger than the target (t(14) = 5.140 p = $1.503\times10^{-4}$), which monotonically reduced to within the target size by the $10^{th}$ trial. In contrast, the LD did not show a monotonic decrease. Unlike in the VDCF, the magnitude of the LD first increased and then decreased. And, again in the de-adaptation phase, we observed that the decays were of opposite sign changes in TE and LD.

To quantify the trajectory adaptation pattern of each group, we performed one-way ANOVAs on the TE and LD values across the trial epochs. The VDCF group showed a significant main effect in LD ($F_{2.546, 35.649}$ = 175.179, p = $3.165\times10^{-5}$, $\eta_p^2$ = 0.926) but not TE ($F_{2.152, 30.134}$ = 2.284, p = 0.116, $\eta_p^2$ = 0.140). Post-hoc Tukey's tests confirmed that the magnitude of LD monotonically changed during the adaptation ($1^{st}$ vs $136^{th}$-$155^{th}$: p<0.001) and de-adaptation ($1^{st}$ vs $131^{st}$-$150^{th}$: p<0.001) phases.

The LIPF group showed a significant main effect in both TE ($F_{1.686, 23.600}$ = 84.204, p = $6.404\times10^{-11}$, $\eta_p^2$ = 0.857) and LD ($F_{2.601, 36.412}$ = 73.312, p = $8.660\times10^{-15}$, $\eta_p^2$ = 0.840). The magnitude of TE monotonically decreased during the adaptation ($1^{st}$ vs $136^{th}$-$155^{th}$: p<0.001) and de-adaptation ($1^{st}$ vs $131^{st}$-$150^{th}$: p<0.001) phases. In contrast, the LD showed a non-monotonic change during the adaptation and de-adaptation phases. The LD increased from the $1^{st}$ to the $3^{rd}$-$5^{th}$ adaptation trials (p<0.001) and then decreased from the $3^{rd}$-$5^{th}$ trials to the $136^{th}$-$155^{th}$ adaptation trials (p<0.001). Similarly, the LD decreased from the $1^{st}$ to $3^{rd}$-$5^{th}$ de-adaptation trials (p<0.001), and then increased from the $3^{rd}$-$5^{th}$ to $131^{st}$-$150^{th}$ de-adaptation trials (p = 0.008).

**The appearance of a new, curved null trajectory after de-adaptation of LIPF.** Furthermore, we observed an intriguing phenomenon in the de-adaptation phase of the LIPF. In the case of the VDCF, upon returning to the null field after the adaptation phase, the participants readily lost their adapted trajectories within the first 10 de-adaptation (null) trials (Fig 2A); their trajectories returned to their original null trajectories (observed in the baseline session) as previously reported [21,22]. This was, however, not the case after the LIPF (see $150^{th}$ de-adaptation trial in Fig 2C). After the de-adaptation phase, the participants' trajectories remained marginally, yet consistently, deviated from their original null trajectories, even after as many as 150 null trials (~20 min). Fig 3A compares the participant-averaged null trajectories before (blue traces) and after (red traces) exposure to the VDCF or LIPF (first and second plots from left). The LD in the null trajectory showed a significant difference between before and after exposure to the LIFP (t(14) = 4.224, p = $8.494\times10^{-4}$), but not the VDCF (t(14) = 0.774, p = 0.452) (Fig 3B).

Crucially, note that the deviation of the new null trajectory was observed to be in the direction in which the force field perturbed the hand and not in the direction opposite to the force field, as would be generally expected after exposure to the VDCF. These observations suggest that the new null trajectory may be not simply an after-effect due to a slow de-adaptation to the force field but a consequence of the TEs induced in the first few null (de-adaptation) trials after exposure to the LIPF. To further investigate the cause of the appearance of the new null trajectory, we next conducted two control experiments.

## Experiment-2

In Experiment-2, we considered the possibility that the new null trajectory was not a consequence of the TE and was, rather, induced due to the LIPF being a position-dependent field.

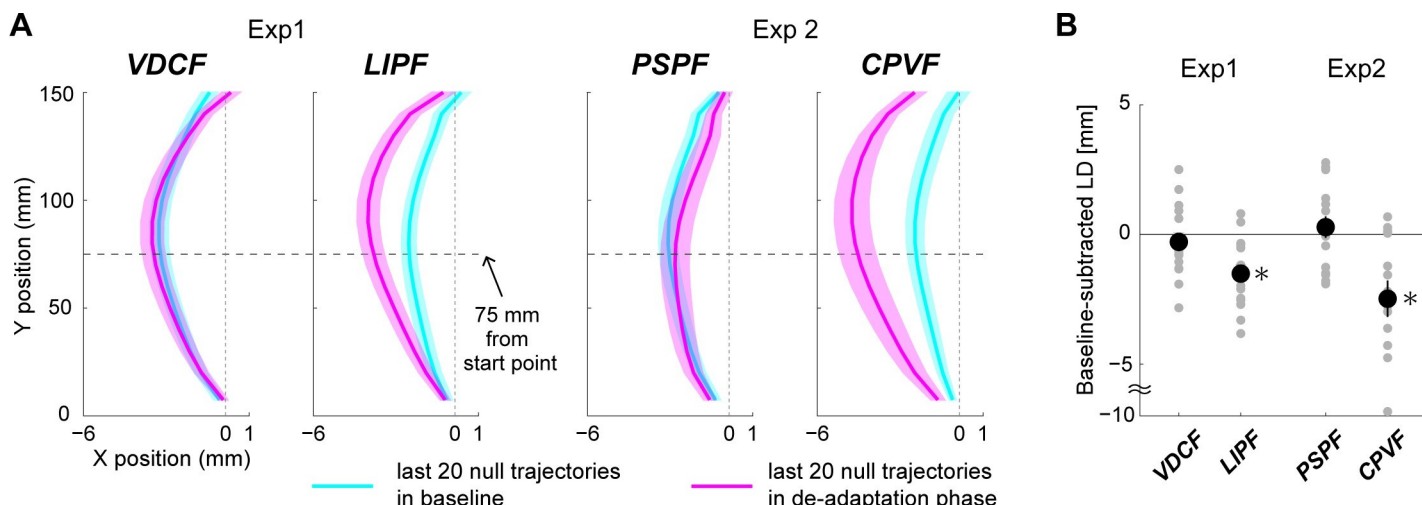

**Fig 3.** Null trajectories before and after adaptation in the four force fields in Experiment-1 (VDCF and LIPF) and Experiment-2 (PSPF and CPVF). A) The null trajectories averaged across the last 20 trials were compared between the baseline (cyan lines) and de-adaptation (magenta lines) phases. The color shades indicate standard errors. Note that the scales differ between x and y axes to clearly show trajectory differences along the x-axis. B) The baseline-subtracted LDs in the trial epoch from the last 20 ($131^{st}$-$150^{th}$) de-adaptation trials in the four force fields. Gray dots represent data from individual participants. The error bars indicate standard errors. * indicates p < 0.05.

To negate this possibility, we examined trajectory adaptation by fifteen participants in the positively skewed position-dependent field (PSPF) (Fig 1B), which is a position-dependent force field that does not induce TEs.

We observed that the magnitude of TE in the first adaptation (t(14) = 0.261 p = 0.798) and de-adaptation trials (t(14) = 0.097 p = 0.924) in PSPF was not significantly larger than the target radius, while the LD showed a monotonic change through the adaptation and de-adaptation phases (see S1 Fig and S1 Text). Importantly, the null trajectory in the de-adaptation phase of the PSPF returned to the baseline null trajectory (t(14) = 0.659, p = 0.520) (Fig 3A and 3B). These observations were similar to the behaviors observed during exposure to the VDCF.

Next, to ensure that the new null trajectory is also observed in other TE-inducing force fields than the LIPF, we examined the trajectory adaptation in the position and velocity-dependent field (CPVF) (Fig 1C). We observed that similar to the LIPF, the CPVF induces a large TE, both in the first adaptation trial (73.2 ± 50.0 (mean ± s.d.) mm, t(13) = 4.905, p = 2.874×10⁻⁴), as well as the first de-adaptation trial (26.0 ± 22.8 mm, t(12) = 5.211 p = 2.178×10⁻⁴). The TEs monotonically reduced until the participant's hand could reach the target. In contrast, as with the LIPF, the LD clearly decreased only after substantial reductions in the TE during the adaptation and de-adaptation phases (see S2 Fig and S1 Text). Crucially, the participants exhibited a new hand trajectory that was significantly different from their initial null trajectory (t(13) = 3.386, p = 0.0049) even after 150 trials in the de-adaptation phase (Fig 3A and 3B). This result provides further support for the possibility that the new null trajectory is a consequence of the TEs induced at the beginning of the de-adaptation phase.

## Experiment-3

Finally, to concretely establish the TEs (at the beginning of the de-adaptation phase) as the cause of the new null trajectory, in Experiment-3 we examined the hand trajectories when the TEs were eliminated in the de-adaptation phase of LIPF (Experiment-1). Thirty participants participated in Experiment-3. Half (15) of these participants had previously participated in Experiment-1. Similar to Experiment 1, these participants trained in the LIPF first, followed by the de-

adaptation phase. However, in the de-adaptation phase of Experiment-3, they made reaches in the null field in the presence of a *partial error clamp* (PEC). This experiment condition was referred to as LIPF-PEC condition, while the LIPF condition of Experiment-1 (the LIPF followed by the Null) was referred to as LIPF-Null condition. The PEC was implemented as a strong spring (see Methods for details) that acted over the second half of their movement ($y > 75$ mm) and pulled the participant's hand to the target along the x-axis (Fig 4A, also see Methods). Note that the first half of the movement ($y \leq 75$ mm), where the LD is measured, remained unaffected by the PEC. The other half of participants, who were newly recruited, experienced the LIPF-PEC first and then the LIPF-Null conditions to cancel out the order effects of these two conditions. We compared the LIPF-PEC condition (Fig 4B, right) with the LIPF-Null condition (Fig 4B, left). As the half of participants was also used in Experiment-1, statistical significance for the data of Experiment-3 was tested with Bonferroni multiple comparison.

Although the trajectory adaptation to the LIPF was similar between the LIPF-Null (left panel in Fig 4B) and LIPF-PEC conditions (right panel in Fig 4B), a stark difference was observed in the de-adaptation phase in presence of the PEC. As expected, the TE in the first PEC trial was substantially attenuated, compared to the first trial in a normal Null field (left panel in Fig 4C; PEC: 8.6 ± 1.9 (mean ± s.d.) mm, Null: 39.9 ± 26.5 mm; $t(29) = 6.550$, $p_{corrected} = 7.133 \times 10^{-7}$). On the other hand, the LD in the first de-adaptation trial did not differ between the PEC field and the Null field ($t(29) = 0.732$, $p_{uncorrected} = 0.470$). However, the difference in LD appeared after the $1^{st}$ de-adaptation trial; while the LD in the LIPF-Null condition showed large jumps from '+x' to '-x', before decaying to the new null trajectory (similar to Experiment-1), the LD in the LIPF-PEC condition was similar to the VDCF condition. In the presence of the PEC, the LD monotonically converged from '+x' through the de-adaptation phase. More importantly, the magnitude of the LD in the last twenty de-adaptation trials in the LIPF-PEC condition was significantly smaller than in the LIPF-Null condition ($t(29) = 2.851$, $p_{corrected} = 0.016$; right panel in Fig 4C). Furthermore, the participants' hand trajectories returned to their initial null trajectories on the application of the PECs ($t(29) = 0.283$, $p_{uncorrected} = 0.779$). Overall, the behaviors in the PEC were observed to be the same as in the no-TE-inducing force fields, specifically the VDCF and PSPF (compare Fig 4B's right panel with Fig 2A). Moreover, when we analyzed only the second half of participants who participated only in Experiment-3 (no need of multiple comparison), we confirmed the same results. The TE in the first de-adaptation trial was substantially smaller in the LIPF-PEC condition than the LIPF-Null condition ($t(14) = 4.193$, $p = 9.020 \times 10^{-4}$). The LD in the last twenty de-adaptation trials was significantly smaller in the LIPF-PEC condition than the LIPF-Null condition ($t(14) = 2.183$, $p = 0.047$), and the hand trajectories in the PEC returned to the original null trajectories ($t(14) = 0.637$, $p = 0.534$). Furthermore, the results of Experiment-3 suggest that muscle fatigue is unlikely to account for the formation of the new null trajectory. This is because we observe new null trajectories in the LIPF-Null but not in the LIPF-PEC conditions, even though the participants train on the same LIPF before performing the de-adaptation phase in these conditions. Overall, these results strongly suggest that the TEs after exposure to TE-inducing force fields caused the new null trajectories observed in Experiment-1 and -2.

## Hierarchy and model simulation

Our results show that in the presence of failure (TE > target size), the evolution of the trajectories is very different from when there are no TEs (compare Fig 2A and 2C). The reduction of TE is consistently given priority over the reduction of LD (Fig 2C), with the TE decreasing monotonically, even at the cost of a temporary increase of LD over several trials. Finally, adaptation in the presence of failure can induce changes in the undisturbed (null) trajectories (Fig 3).

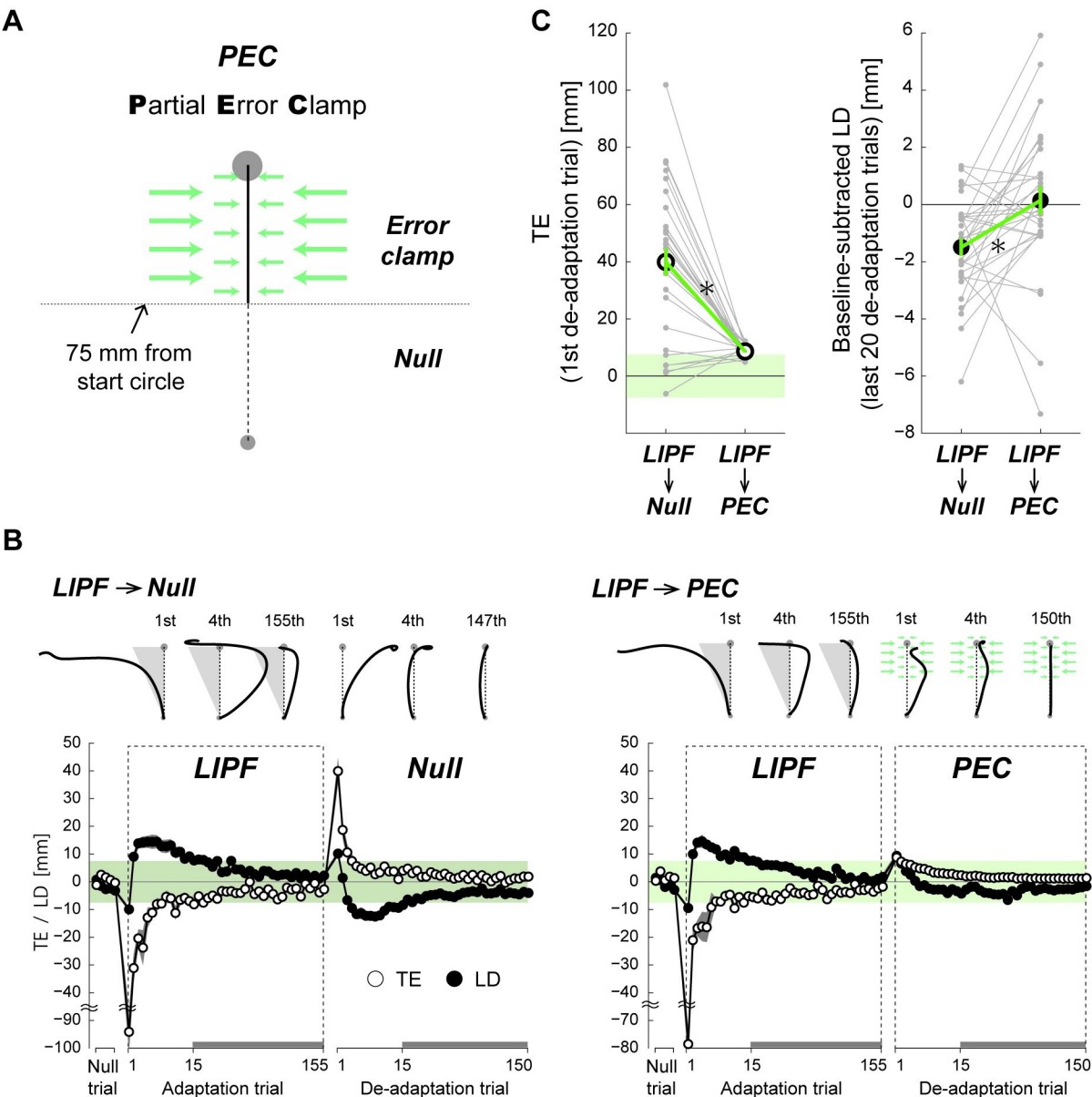

**Fig 4. Effect of attenuation of TE on the de-adaptation trajectory.** (A) After exposure to LIPF, the participants in the LIPF-PEC condition of Experiment-3 were exposed to the PEC where a force channel was applied over the second half of the reaching movement to attenuate TEs. (B) The hand trajectories and learning curves of both TE (open circle) and LD (filled circle) are compared between the LIPF-Null (left panel) and LIPF-PEC conditions (right panel). (C) The TE in the first de-adaptation trial (left panel) and the baseline-subtracted LD averaged across the last 20 (131st-150th) de-adaptation trials (right panel) were compared between the two conditions. Gray dots represent data from individual participants. The error bars indicate standard errors. * P < 0.05.

First, these observations suggest the presence of a TE-driven adaptation process, in addition to the SPE-driven internal model adaptation. Furthermore, the distinct adaptation of the TE and LD in the LIPF, one of which is monotonic while the other not (Fig 2C), led us to hypothesize a hierarchical interaction between the two processes. To evaluate this hypothesis, we simulated the trajectory adaptation in the VDCF, LIPF-Null, and LIPF-PEC using two sensorimotor adaptation models that consider only the internal model adaptation, with and without the addition of a hierarchical TE-driven adaptation process.

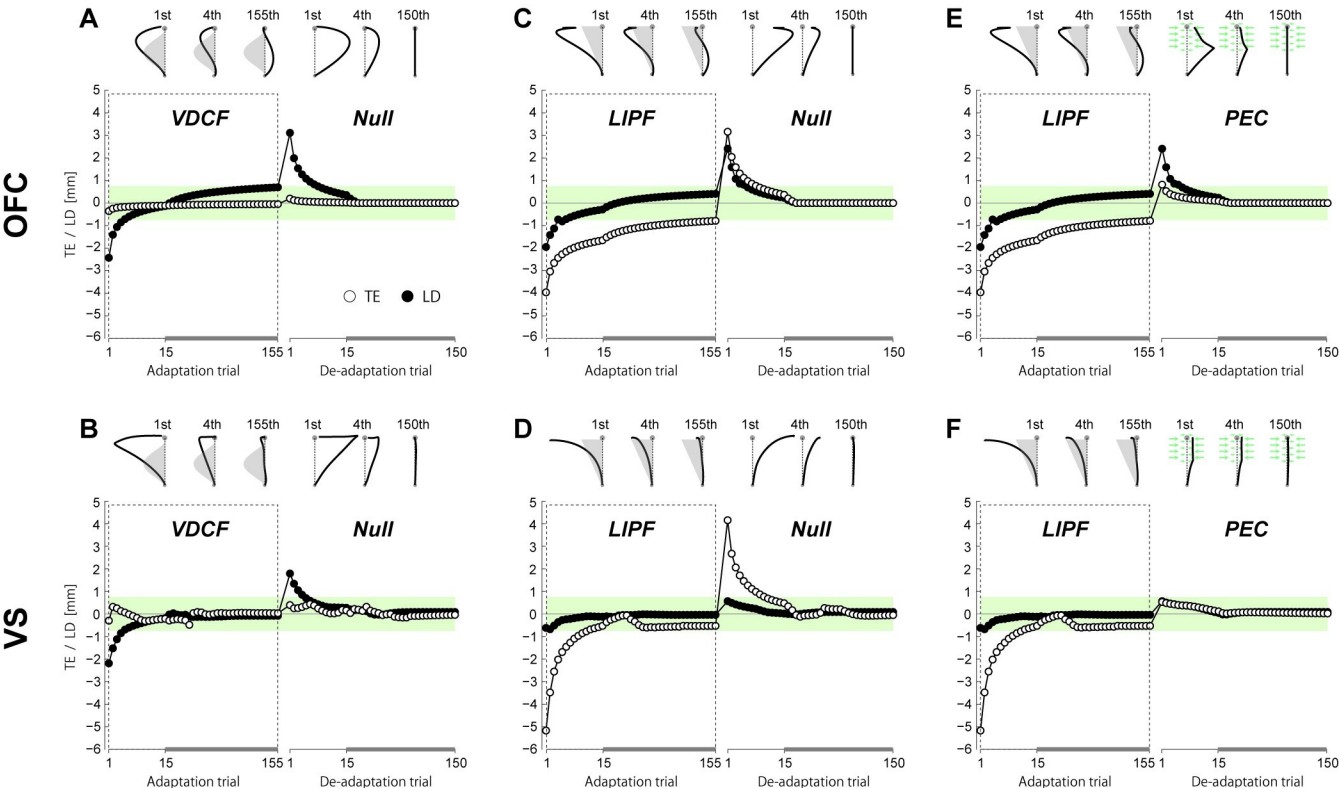

**Fig 5.** Flat models cannot reproduce LIPF and PEC behaviors: Simulation for trajectory adaptation in the VDCF (A, B), LIPF-Null (C, D), and LIPF-PEC (E, F) conditions, represented by TE (open circle) and LD (filled circle) by the flat OFC (upper panels) and VS models (lower panels). The flat learning models (only internal model adaptation) were unable to reproduce either the non-monotonic change in LD (C, D, E, F) or the curved null trajectory with a persistent deviation after exposure to the LIPF(C, D).

First, we started with the 'flat' optimal feedback control model (or the flat OFC model), proposed by Izawa et al. [23] to explain trajectory adaptation in a velocity-dependent force field by combining the internal model learning of the learned force field and the optimal feedback control [24]. Second, the 'flat' V-shaped model (or flat VS model) proposed by Franklin et al. [25], which utilized a different algorithm, similar to feedback error learning [6] where muscle activation changes across trials are determined by a V-shaped learning function under the assumption of a pre-planned desired trajectory. We refer to both these models using the prefix 'flat' as both models consider a single SPE-driven internal model adaptation process to explain motor adaptations. We will show that these models can explain our experimental observations by appending a 'hierarchical' TE-driven adaptation process in their current structure. Please see Methods for details of implementation.

Fig 5 shows that simulations of the VDCF, LIPF-Null, and LIPF-PEC adaptations by the flat OFC and flat VS models. Although the flat OFC model (Fig 5A) and the flat VS model (Fig 5B) qualitatively reproduced the trajectory adaptation in the VDCF well, they were unable to reproduce both the non-monotonic change in LD and the persistent curved null trajectory observed in the LIPF-Null and LIPF-PEC (Fig 5C and 5D).

Next, we introduced an additional TE-driven adaptation process to these models. We assumed that the adaptation process represents a modification of the kinematic plan, when there is a failure (i.e., a TE > target size), and then added the kinematic plan adaptation process on the top of the flat learning models (Fig 6A). We thus refer to these two models as the

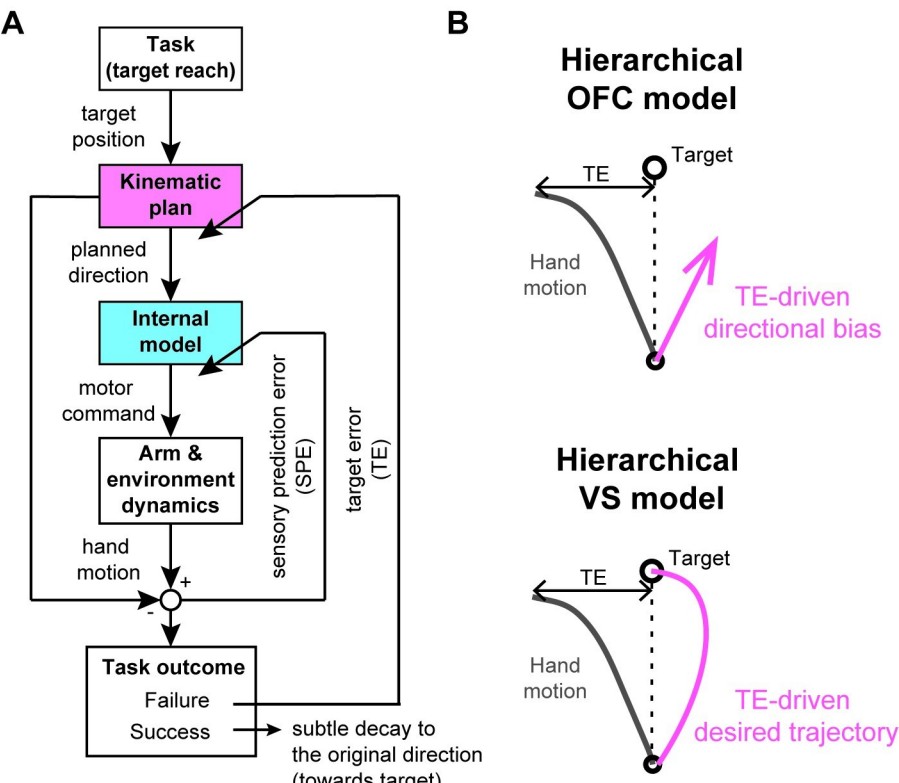

**Fig 6. Hierarchical motor adaptation model.** (A) Schematic diagram of the model. The model consists of two adaptation components: the kinematic plan adaptation (magenta box) as a higher component, driven by TE, and the internal model adaptation (light blue box) as a lower component, driven by SPE. In the presence of failure (i.e., TE > target size), the kinematic plan adaptation process becomes active and modifies the planned direction of the hand motion. When the task is successful, the planned direction slowly decays to the original movement direction. (B) The planned direction of the hand motion is implemented as a directional bias (magenta arrow) in the hierarchical OFC model and a desired trajectory (magenta line) in the hierarchical VS model (see Methods for details).

'hierarchical' OFC model and the 'hierarchical' VS model, respectively. The kinematic plan adaptation process was assumed to be activated only in the presence of failure and modulated by TE so that the trajectory is adjusted to change in the opposite direction to the TE. In the absence of failure (i.e., TE < target size), the kinematic plan subtly decays across trials to the original plan (i.e., the straight direction towards the target). We assume that the decay stops when the motor cost of the generated reaching goes below a small value of threshold (see Methods for details of implementation). This assumption was done to reproduce the persistent curved null trajectory.

In the hierarchical OFC model, this process was implemented by a direction bias [26] (Fig 6B), which was incorporated into the cost function within the flat OFC model (see Methods for details). In the hierarchical VS model, the initial direction of the desired trajectory (Fig 6B) was modified in the same way as the hierarchical OFC model (see Methods). By including this TE-driven adaptation process, both models (Fig 7C, 7D, 7E and 7F) could explain all the features of the trajectory adaptation in LIPF-Null and LIPF-PEC, including the non-monotonic change of the LD during the adaptation phase, and the appearance of the new null trajectory after de-adaptation in the LIPF-Null or disappearance of that in the LIPF-PEC. In the absence of failure, as in VDCF, both models predict the same results as their flat counterparts (Fig 5A and 5B).

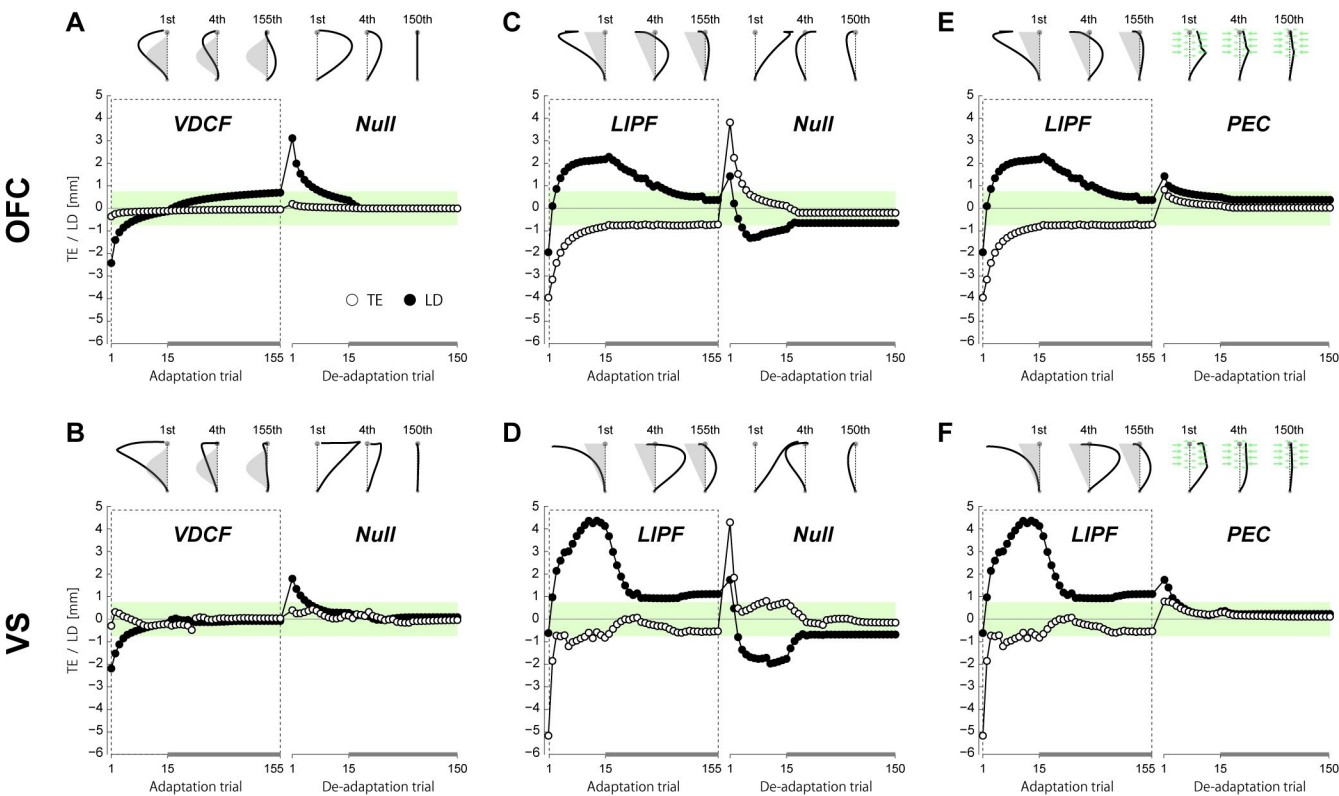

**Fig 7.** Hierarchical model's simulation for trajectory adaptation in the VDCF (A, B), LIPF-Null (C, D), and LIPF-PEC (E, F) conditions, represented by TE (open circle) and LD (filled circle) by the hierarchical OFC (upper panels) and VS models (lower panels). The simulated hand trajectories were shown at the top of each panel. The hierarchical learning models (kinematic plan adaptation and internal model adaptation) successfully reproduced the behaviors in all the three conditions.

## Discussion

We examined the motor adaptation of arm reaching trajectories in force fields that induce failure (TE > target size) at the beginning of the adaptation and de-adaptation phases. First, our results showed that the human motor learning system puts a higher priority on the reduction of TE than LD. In the presence of failure, the LDs did not follow a typical monotonic decrease as reported in previous studies [21,22,27,28]. TE is reduced first, even at the expense of an increased LD (Fig 2C). A monotonic decrease in LD took place only after the TE was reduced to around the target size. Second, the presence of failure in the de-adaptation phase caused the appearance of a new null trajectory that was distinct from the null trajectory observed in the baseline period and persisted even after 150 de-adaptation trials. These observations were successfully reproduced by the hierarchical motor adaptation models that combine a TE-driven kinematic plan adaptation with the internal model adaptation.

The prioritized reduction in TE over LD (Fig 2C and 2D) cannot be explained only by internal model adaptation even when considering multiple time scale adaptations, such as a two-state model [10–12], because these models predict similar monotonic changes in both TE and LD (like Fig 5). It is important to note that this is also the case when considering the spatiotemporal difference in the error information. If the errors early in a trajectory are less important than those at the end to update the internal model of the force field, the difference may affect the adaptation rate (i.e. TE may lead to a faster internal model adaptation) but still not change the adaptation pattern to which the internal model adaptation leads (i.e. monotonic decay of

the trajectory). In contrast, the non-monotonic trajectory changes in the presence of TEs suggests the presence of an additional TE-driven kinematic plan adaptation. In our hierarchical motor adaptation models (Fig 6), the kinematic plan adaptation changes the reaching direction in the opposite direction of the TE, which enables a quick reduction in TE, even when it sometimes leads to an increase in LD (Fig 7C, 7D, 7E and 7F). After the TE reduction, we assume that the kinematic plan slowly returns towards the original movement direction (i.e., towards the target). The hierarchical addition of this TE or failure driven process enables the models to explain the TE and LD adaptation processes both in no-TE-inducing force-fields as well as TE-inducing force fields.

The appearance of the new null trajectory in the de-adaptation phase can be also explained by the hierarchical dominance of kinematic plan adaptation over internal model adaptation. In our hierarchical models, we assumed that after the motor cost of arm reaching falls below a small threshold value, the decay of the kinematic plan toward the baseline plan stops. This assumption could reproduce the persistent curved null trajectory after de-adaptation in the presence of failure. The models thus suggest that the TE-driven kinematic plan adaptation may determine the steady-state null trajectory to which the internal model adaptation converges. This possibility is strongly supported by Experiment-3 (Fig 4) where the suppression of TE enabled the participants to converge back to their baseline null trajectory. This observation was also successfully reproduced by the hierarchical learning models (Fig 7E and 7F). Our assumption, that the TE-driven kinematic plan adaptation is also affected by the motor cost, is similar to the idea that a desired trajectory of movement may be modified according to the level of interaction force with the environment [29]. It has however not yet been empirically examined and remains an interesting question for future studies.

Motor learning processes like motor memory [30–32] or use-dependent learning [33] make one's movement similar to the last performed movement. Operant reinforcement learning [34] causes people to select movements for which the task had previously been successfully achieved. These processes may be seen as likely candidates to explain the persistent curved trajectories. However, these processes alone cannot explain why the persistent curved null trajectories do not appear in the no-TE-inducing force fields (VDCF or PSPF) (Fig 3B) as well as during PEC in Experiment-3 (Fig 4B), in which the participants successfully reached the target with curved null trajectories in the first de-adaptation trials. Our results thus suggest that even if motor memory, use-dependent learning, or operant reinforcement learning is indeed active during the force field adaptation, unlike kinematic plan adaptation, they do not hierarchically interact with internal model adaptation but instead work in a non-hierarchical manner. Likewise, other possible causes like perceptual bias [35,36] or perceptual recalibration [37] of the hand position also can not well explain why the persistent curved null trajectories appear only in the TE-inducing but not the non-TE-inducing force fields. If our model includes these learning processes or perceptual adaptation processes, it may be able to better explain the behavior. We however note that the main purpose of our model simulation is to explain the necessity of an additional TE-driven adaptation process hierarchically interacting with the internal model adaptation, rather than develop a new model. For this purpose, we chose the two most popular learning models in the current literature and demonstrated the effect of adding the additional TE driven process.

A priori, the new null trajectory in the de-adaptation phase shown by us is different from a persistent retention of learned movements that has been recently reported to occur after some reinforcement period where only binary success or failure feedback is provided [38–41]. There, the retention was measured in no feedback periods subsequent to reaches, in which movement-related feedback was not available, and then when the feedback was available, a typical washout process took place with the movement quickly returning to the baseline level

[38]. In contrast, in our study the new null trajectory persists for at least ~20 min of the de-adaptation period even when movement-feedback is available and without a reinforcement period. Future work is needed to determine how long the new null trajectory persists or whether it decays very slowly.

Recent studies have identified the presence of distinct explicit and implicit components of adaptation to novel visuomotor rotations [7,9,10]. The explicit components, called explicit strategy learning, have been proposed to be sensitive to task performance or TE, and faster than implicit components represented by internal model adaptation. We believe the TE-driven adaptation process we observed here may (at least partially) be an explicit strategy learning, as it was active only in the presence of failures and fast [10,14] but insensitive to LD (i.e. SPE). However, the key difference between this TE-driven adaptation and the explicit strategy learning previously identified lies in the way the two processes interact with the internal model adaptation. Previous visuomotor rotation studies have often utilized a two-state model to explain the interaction between the explicit strategy adaptation and internal model adaptation [10,42] by assuming that these two adaptation processes interact in a non-hierarchical manner where the net reach trajectory is defined to be the sum of the two. However, in the case of visuomotor rotation tasks, the parameter to be learned by the two adaptation processes is the same–the rotation angle (or its equivalent). In fact, previous force-field studies have similarly looked at the adaptation of a single parameter–the trajectory (quantified by its curvature, deviation, or encompassed area relative to the straight line). The adaptation of a single parameter is well explained by 'flat' models, including the "non-hierarchical" two-state model. On the other hand, this is not the case in our force field task, where the two adaptation processes represent changes in distinct parameters (the target and trajectory). The net adaptation behavior in our experiment cannot be explained by the flat models, including the two-state model, in its current formulation. Rather, the TE-driven adaptation and the internal model adaptation we observe here seem to be more consistent with the traditional view of hierarchical motor planning of kinematics and dynamics [6,20].

Our hierarchical models are different from the Adaptation Modulation model, a hierarchical motor adaptation model proposed by Kim et al. [15] that could explain the interaction of TE-driven adaptation and SPE-driven adaptation in their visuomotor rotation paradigm. The Adaptation Modulation model increases the adaptation rate of the SPE-driven adaptation process in the presence of failure (TE > target size). In the end, as with the two-state model, this model also considers only adaptation of the internal model (i.e. novel visuomotor ration) although it is modulated by the presence of TE. Thus, the TE-driven process of the Adaptation Modulation model hierarchically determines a temporal feature of the SPE-driven adaptation (i.e., how fast arm trajectories adapts to the novel environment) but not a spatial feature as in our models (i.e., where the adapted trajectories converges). Accordingly, the Adaptation Modulation model can explain our observations in the non-TE-inducing fields but not those in the TE-inducing fields (i.e. the presence of non-monotonic trajectory change and new null trajectory). Additionally, this is also true for two other models proposed by Kim et al. [15]: the Movement Reinforcement and Dual Error models. Both models implement an interaction of SPE-driven and TE-driven processes, but again consider the adaptation of internal model alone. On the other hand, our model may partially explain their results. Specifically, the TE-driven kinematic plan (by limiting the range of the plan change) can explain the facilitated adaptation observed in Kim et al. [15] although some constraints are necessary. However, as the TE-driven process in Kim et al. [15] modulates adaptive behavior in a completely implicit manner while our TE-driven process may, we believe, work in an explicit manner, these two may be distinct in nature.

Studies have regularly found hierarchical behaviors during cognitive learning and decision making in humans. The brain activations during these hierarchical behaviors have been well explained by hierarchical reinforcement learning (HRL) algorithms [43–48]. The typical role of the higher component in a HRL system is to select a task-goal-oriented sub-goal or option, while the lower component typically selects an action to achieve this goal or sub-goal [44,49–51]. This structure is very similar to the hierarchical motor learning models we suggest here. However, while the previous theoretical and imaging studies have exhibited a hierarchy at the level of cognitive learning in low degrees-of-freedom tasks, here our study suggests the presence of similar hierarchical structures for solving large degrees-of-freedom motor learning problems. The higher components active during cognitive learning have been linked to neural systems in the dorsolateral striatum, the dorsolateral prefrontal cortex, the supplementary motor area, the pre-supplementary motor area, and the premotor cortex [44]. On the other hand, the lower components have been related to the ventral striatum and the orbitofrontal cortex that has strong connections to both the ventral striatum and the dorsolateral prefrontal cortex [44]. Interestingly most of these areas have been observed to be active during motor learning of point-to-point arm or finger movements as well [52–55], suggesting the cognitive learning processes and the hierarchical motor learning may process as subsets of a common HRL structure. However, further studies are required to clarify this speculation by concretely examining the sharing of neural structures between the two processes.

Before the conclusion, we note two limitations of this study. First, while we manipulate the presence or absence of TE across the force fields, the current experimental design could not control several movement features like the velocity profile, stiffness profile [56], online feedback gain [57], posture at the final position [58] or adaptive movement changes [59,60]. Although we believe it is unlikely that any of these factors alone can consistently explain our two key observations in the TE-inducing force fields: the non-monotonic trajectory change and the new null trajectory, they may partially contribute to the formation of our observations. For example, one possibility is that a change in feedback gain induced by a large TE may contribute to shape the new null trajectory, because feedback control has been suggested to share the internal model used for 'feedforward' control [61–63]. Another possibility is that the faster reduction of TE than LD may be boosted by adaptive control that involves online update of the control policy within individual movements [59,60]. Adaptive control may update not only the control policy but also the kinematic plan in the presence of TE. If the update rate to the kinematic plan is greater than that to the control policy, this may result in different trial-by-trial adaptation of TE and LD with faster and slower time scales, respectively. Future studies are needed to examine these possibilities.

Second, the current experiment design cannot determine whether the TE-driven kinematic plan adaptation is an implicit process to automatically compensate for TE or an explicit process to intentionally change the strategy or the initial reach direction, although we believe the latter. One promising way to address this question may be to manipulate the participants' psychological sensitivity to TE of the same movements as employed by Kim et al. [15]. Changing the target size or monetary reward for task success, but with other motor factors being kept constant, would be useful to examine whether or not the TE-induced adaptative behaviors observed in our study are explicitly modulated.

The failure (i.e., TE) driven adaptation of the kinematic plan leads to large and fast movement changes that are arguably costly in terms of control and energy [24,64]. It is, therefore, possible that in our daily lives, to reduce the control cost, kinematic plan adaptation remains inactive during the performance of most movements, as they are overlearned and rarely lead to failure. This plan adaptation is likely activated only when there is a (probably unexpected) failure. When a failure is experienced, the kinematic plan adaptation process helps the brain to quickly acquire success or reward, even at the expense of large high energy movement changes,

after which it is again left to the internal model adaptation to optimize the movement relative to this new movement plan. Furthermore, task success or failure definitively depends on task requirements. In our study, as TE determines whether the task is successful or not, the participants prioritized TE over LD. However, if participants were instructed that the task goal is to make a reaching trajectory with a certain magnitude of LD, they would prioritize LD over TE. Moreover, when the failure is indicated by a binary (success or failure) feedback but not a signed error feedback like TE, LD may be more prioritized as suggested in a previous study [65]. Importantly, whatever the task goal or the feedback type is, our results suggest that the presence of failure may activate the kinematic plan adaptation to quickly achieve the goal. In conclusion, our study provides behavioral evidence to exhibit that human motor learning is shaped by the hierarchical interactions between the two learning processes; a *higher* kinematic plan adaptation driven by failure, and a *lower* internal model adaptation. This hierarchical motor adaptation structure may allow the brain to negotiate unexpected behavioral failures in an ever-changing and diverse environment around us.

## Methods

### Ethics statement

All experiments involved human participants and were approved by both the ethics committees of Advanced Telecommunication Research Institute (approval numbers: 15–722, 16–722) and National Institute of Information and Communications Technology. All participants signed an institutionally approved consent form.

### Participants

A total of seventy-five neurologically normal volunteers (fourteen females and sixty-one males; age 22.70 ± 2.06, mean ± s.d.) participated in the experiments. All participants were right-handed as assessed by the Edinburgh Handedness Inventory [66]. All participants were naïve to the purpose of the experiments. No statistical methods were used to determine sample sizes although the sample sizes used in this study were similar to those in previous studies using similar reaching tasks [9,14,15,23,42].

### Apparatus

The participants sat on an adjustable chair while using their right hand to grasp a robotic handle of the twin visuomotor and haptic interface system (TVINS) used to generate the environmental dynamics [67]. Their forearm was secured to a support beam in the horizontal plane and the beam was coupled to the handle. Since the TVINS has two parallel-link direct drive air magnet floating manipulandums, we performed the experiments with two participants at a time. Each manipulandum was powered by two DC direct-drive motors controlled at 2,000 Hz and the participants' hand position and velocity were measured using optical joint position sensors (4800,000 pulses/rev). The handle was supported by a frictionless air magnet floating mechanism.

A projector was used to display the position of the handle with an open circle cursor (diameter 4 mm) on a horizontal screen board placed above the participant's arm. The screen board prevented the participants from directly seeing their arm and handle. The participants controlled the cursor representing the hand position by making forward reaching movements (the details will be shown in the next section) from a start circle (10 mm diameter) to a target circle (15 mm diameter), which were displayed on the screen throughout all of the experiments. The start circle was located approximately 350 mm in front of the shoulder joint, and the target was 150 mm away from it.

## Task

The participants were instructed to move the cursor from the start circle to the target circle in a period of 400 ± 50 ms. No instructions were given about the trajectory of reaching movement. Each movement was initiated by audio beeps. Participants were instructed to begin movement on the second beep, 1 s after the first beep. The second beep lasted for 400 ms and could be used as a reference to the instructed movement duration. The cursor was visible only during each trial. After each trial, the participants were provided information about their movement duration and final hand position. Movement duration was defined as the period between the time the cursor exits the start circle and enters the target circle. Participants were provided information about the movement duration, given as "SHORT", "LONG" or "OK". The final hand position was defined as the position at the moment when the hand velocity fell below 20 mm/s. If the final hand position was within the target circle, the inside of the circle turned blue. After each trial, a third beep 3s after the first beep indicated the termination of the trial and the TVINS brought the participant's hand back to the start circle, and the next trial started after a period of 1 s. The inter trial-interval was 8 s.

## Force fields

This study used four different force fields: Velocity-dependent curl field (VDCF), Linearly increasing position-dependent (orthogonal) field (LIPF), Positive skew position-dependent (orthogonal) field (PSPF), and Combination of position- and velocity-dependent field (CPVF). There are two TE-inducing force fields (LIPF and CPVF) and two no-TE-inducing force fields (VDCF and PSPF). They are illustrated in Fig 1C and computed using the following equations.

$$\text{VDCF}: \begin{bmatrix} F_x \\ F_y \end{bmatrix} = B_1 \begin{bmatrix} 0 & -1 \\ 1 & 0 \end{bmatrix} \begin{bmatrix} \dot{x} \\ \dot{y} \end{bmatrix}$$

$$\text{LIPF}: \begin{bmatrix} F_x \\ F_y \end{bmatrix} = K_1 \begin{bmatrix} 0 & -1 \\ 0 & 0 \end{bmatrix} \begin{bmatrix} x \\ y \end{bmatrix}$$

$$\text{PSPF}: \begin{bmatrix} F_x \\ F_y \end{bmatrix} = K_2 \frac{cos(\pi + 40y) + 1}{(\pi + 40y)^5} \begin{bmatrix} -1 \\ 0 \end{bmatrix}$$

$$\text{CPVF}: \begin{bmatrix} F_x \\ F_y \end{bmatrix} = K_1 \begin{bmatrix} 0 & -1 \\ 0 & 0 \end{bmatrix} \begin{bmatrix} x \\ y \end{bmatrix} - B_1 \begin{bmatrix} 0 & -1 \\ 1 & 0 \end{bmatrix} \begin{bmatrix} \dot{x} \\ \dot{y} \end{bmatrix}$$

Where $(F_x, F_y)^T$ represents a force in Newtons exerted on the hand, $(x, y)$ is the hand position relative to the center of the start circle in meters, $(\dot{x}, \dot{y})$ is the hand velocity in meter per second, $B_1$ is 14 Ns/m, $K_1$ and $K_2$ are 60 and 20868 N/m, respectively.

Importantly, the hand motion is momentarily constrained to the final hand position where the velocity fell below a low threshold of 20 mm/s by applying a strong stiff two-dimensional spring force (500 N/m) and damper (50 Ns/m). The constraint force is active until the trial ends (lasting for around 1600 ms). This was designed such that participants did not need to continue resisting large force at the movement end (as in LIPF and CPVF) and it prevents them from reaching the target by sub-movements [68,69].

## Partial error clamp

This study developed a new error clamp method and used it in Experiment-3. Previous motor learning studies have extensively utilized error clamp methods to assess motor adaptation performance [70]. When the error-clamp was active, the trajectory of the hand was attracted to a straight line joining the start circle to the target by a virtual "channel" (see Fig 4A) in which any motion perpendicular to the straight line was pulled back by a one-dimensional spring (800 N/m) and damper (45 Ns/m). However, in contrast to the previous experiments, the error clamp was applied only over the last part of the hand movement (y >75 mm) such that the first part of the movement where the LD is measured (the details will be shown in a later section) is unaffected by the clamp. Furthermore, the magnitude of the spring was set weaker than that in the previous studies, which allows the hand trajectory to change smoothly (see the hand trajectories for LIPF-PEC condition in Fig 4B). We call this a partial error clamp (PEC).

## Experiment procedure

**Experiment-1.** Thirty participants who passed initial screening (the details will be shown later in *Participant screening* section) were randomly assigned to each of the two groups (n = 15 for each): the VDCF group and the LIPF group (Fig 1C). First, the participants in both groups were given a practice period to acclimatize themselves to the apparatus and task. They were allowed to take their time but asked to make reaching movements in the no-force field environment (null field) at least more than 50 trials. All participants finished practice less than 100 trials. This was followed by the two experimental sessions: baseline and adaptation sessions. In the baseline session, the participants performed 50 trials of reaching movements in the null field. In the adaptation session, after 5 trials in the null field, the participants in the VDCF and LIPF groups performed 155 (adaptation) trials in VDCF and LIPF, respectively, which was followed by 150 (de-adaptation) trials in the null field. Two-minutes rests were taken three times, each after the $50^{th}$, $100^{th}$, and $150^{th}$ adaptation trials.

**Experiment-2.** Thirty participants who passed initial screening were randomly assigned to each of the two groups (n = 15 for each): the PSPF group and the CPVF group (Fig 1C). The experimental procedure is the same as Experiment-1.

**Experiment-3.** Thirty participants took part in Experiment-3. Half of them who were assigned to the LIPF group of Experiment-1 returned to our laboratory at least more than one week after Experiment-1 and performed Experiment-3. In Experiment-3, unlike Experiment-1, they performed 155 adaptation trials in the LIPF followed by 150 de-adaptation trials in the PEC. Thus, this experimental condition was referred to as the LIPF-PEC condition, while the condition in the Experiment-1 performed by the participants was called as LIPF-Null condition. To compare these two conditions, we needed to cancel out the order effects of the two experimental conditions. We thus newly recruited another fifteen participants. Those who passed initial screening experienced the LIPF-PEC first and then the LIPF-Null conditions. These experiments in the two conditions were again separated by at least one week. The experimental procedure in Experiment-3 is also the same as Experiment-1 except that in the LIPF-PEC condition, the participants performed the 155 de-adaptation trials in the PEC.

## Data analysis

Target error (TE) and lateral deviation (LD) were used to evaluate motor adaptation. The TE was defined as x-deviation of the final hand position from the straight line joining the start circle to the target (Fig 1B). The final hand position was defined as the position at the moment when the hand velocity fell below 20 mm/s. The LD was defined as the x-deviations midway (at 75 mm from the start circle) from the straight line joining the start circle to the target.

To draw the participant-averaged trajectories for each of the VDCF (Experiment-1), LIPF (Experiment-1), PSPF (Experiment-2), and CPVF (Experiment-2) conditions, we sampled the x-axis data at the fifteen y positions: 7.5 (target radius size), 10, 20, 30, 40, 50, 60, 70, 80, 90, 100, 110, 120, 130, 140, and 150 (target position) mm for each participant. These sampled data were averaged across participants for each y position and plotted in Fig 3A.

All statistical tests conducted in this study were two-tailed with a significance level of 0.05. To examine changes in each of TE and LD during motor adaptation, we separately performed one-way ANOVAs across trial epochs (6 epochs:1st, 3rd-5th, 136th-155th adaptation trials and 1st, 3rd-5th, 131st-150th de-adaptation trials). When assumptions of heterogeneity of covariance were violated, the number of degrees of freedom was corrected with the Greenhouse-Geisser procedure. Post-hoc pairwise comparisons were performed using Tukey's method. For other tests, we performed paired or unpaired t-test was performed. The ANOVAs were performed using SPSS Statistics ver. 25 (IBM) and the t-tests were performed using MATLAB version R2018b (Mathworks).

## Data exclusion

Trials were excluded from the analysis when the reach distance was less than 75 mm as the LD could not be evaluated (Fig 1B). 34 trials (0.098% of the total number of trials) were excluded. Only one participant in the CPVF group was excluded from the analysis because the participant showed unstable trajectory changes over the last 100 de-adaptation trials with at least three large jumps ($> 20$ mm) across the x-axis as well as an outlying value of the LD over the last 20-de-adaptation trials (outside of 3 s.d. from the mean). 14 participants were thus analyzed for the CPVF group (Experiment-2). Note that for the t-test on the first de-adaptation trial of the CPVF group, the statistical degree of freedom was 12 since the first de-adaptation trial of a participant was excluded due to the trial exclusion criterion.

## Participant screening

We screened participants in all the experiments based on trajectory deviation in the baseline sessions. With pilot experiments, we anticipated the persisting curved null trajectory would appear after adaptation to TE-inducing force fields as seen in Fig 3. To assess this phenomenon, we wanted to examine how much the curved trajectories differ from null trajectories in the baseline. However, our pilot experiments observed that some participants showed considerably curved null trajectories (LD of $\sim 10$ mm) in the baseline session because we did not provide participants with any instruction on reaching trajectory for the sake of the research question. Thus, to ensure that baseline null trajectories are the same across all the participant groups, only the participants whose LD averaged over the last 20 trials in the baseline session is less than 4.5 mm proceeded to the learning session. In fact, there were no significant differences in the LD in the baseline session across all the groups: the VDCF, LIPF, PSPF, LIPF groups and the participant group who performed the PEC-LIPF condition first (one-way ANOVA: $F(4, 73) = 1.430$, $p = 0.223$, $\eta_p^2 = 0.077$). The other screened out participants afterwards performed similar reaching experiments which is not related to this study, and thus their data were not further analyzed for this study.

## Simulation

To explain adaptive behaviors in the VDCF and the LIPF of the Experiment-1, we utilized two motor learning models: one is proposed by Izawa et al. [23], which we refer to as the flat OFC model, and the other is proposed by Franklin et al. [25], which we refer to as the flat VS model. These original models implement only the internal model learning and can explain monotonic

trajectory adaptation as observed in the VDCF. However, they cannot explain non-monotonic trajectory adaptation, nor a persistent change in the null trajectory in the LIPF. We thus extended the two models by introducing a TE-driven kinematic plan adaptation that hierarchically interacts with the internal model adaptation (Fig 6A). We referred to the extended models as the hierarchical OFC model and the hierarchical VS model, respectively.

**OFC model.** The original model (i.e., flat OFC model) utilizes optimal feedback control (OFC) theory [24,71] to simulate reaching trajectories during adaptation to a state-dependent novel force field, based on a concept that motor learning is a process to acquire a model of the novel environment and use the model to re-optimize movements. Accordingly, in this framework, motor adaptation is characterized by the knowledge of the environment (the novel force field) which the motor system gradually acquires. The external force imposing to the arm is written by the form:

$$F_t = Dx_t \tag{1}$$

where $\mathbf{F}_t$ and $\mathbf{X}_t$ are the external force vector and the current state vector of the plant (arm and environment) at time $t$, respectively. $\mathbf{D}$ is the force matrix (e.g. for VDCF, $\mathbf{D} = B_1[0{-}1;1\ 0]$). What the motor system needs to perform the optimal movement in the force field is the full knowledge of $\mathbf{D}$, which is assumed to be gradually acquired. The knowledge of $D$ during adaptation is represented by the form:

$$\widehat{D} = \alpha D \tag{2}$$

where $\widehat{D}$ is the estimated force matrix, and $\alpha$ is the learning parameter, which is assumed to gradually increase from 0 to 1 with adaptation. During adaptation, the motor system predicted the external force using $\widehat{D}$ as follows:

$$\widehat{F}_t = \widehat{D}\widehat{x}_t \tag{3}$$

$\widehat{F}_t$ is the predicted external force vector at time $t$ and $\widehat{x}_t$ is the estimated state vector of the plant and is obtained through the optimal state estimator (see [71]). Accordingly, the motor system produces the motor command optimized for the environment where the predicted external force could impose on the arm. Only when $\alpha = 1$, does the system have the full knowledge of $D$ and produce the optimal motor commands for the actual environment. When $0 < \alpha < 1$, the system has an incomplete knowledge of $D$ and would produce a sub-optimal movement for the actual environment. Thus, by changing the value of $\alpha$, Izawa et al. simulated reaching trajectories in several phases of motor adaptation using OFC.

For the hierarchical OFC model, we borrowed the idea of a kinematic bias of movement direction proposed by Mistry et al. [26], which we refer to as directional bias. Mistry et al. extended the cost function of OFC by including a directional bias to explain a directional preference of reaching trajectories observed during motor adaptation to an acceleration-based force field. The directional bias represents the desired direction of movement, which is represented by the form:

$$Q_d = \begin{bmatrix} d_y^2 & -d_x d_y \\ -d_x d_y & d_x^2 \end{bmatrix} \tag{4}$$

Where $\mathbf{Q}_d$ is the directional bias matrix and $\boldsymbol{d} = [d_x\ d_y]^T$ is the desired directional vector represented as a unit vector. While the original cost function consists of the error term (first term) and the motor cost term (second term) (Eq 5), the cost function of the hierarchical OFC model (Eq 6) has the additional term related to the directional bias (third term in Eq 6) so that

any position or velocity perpendicular to the desired direction was penalized as follows:

$$\boldsymbol{x}_t^T \boldsymbol{Q}_t \boldsymbol{x}_t + \boldsymbol{u}_t^T \boldsymbol{R} \boldsymbol{u}_t \tag{5}$$

$$\boldsymbol{x}_t^T \boldsymbol{Q}_t \boldsymbol{x}_t + \boldsymbol{u}_t^T \boldsymbol{R} \boldsymbol{u}_t + e^{-t/\tau}(k_p \boldsymbol{p}_t^T \boldsymbol{Q}_d \boldsymbol{p}_t + k_v \boldsymbol{v}_t^T \boldsymbol{Q}_d \boldsymbol{v}_t) \tag{6}$$

where $\boldsymbol{x_t}$ is the current state vector of the plant (the arm and environment) at time $t$, $\boldsymbol{u_t}$ is the motor command vector, $\boldsymbol{p_t}$ and $\boldsymbol{v_t}$ are the position and velocity vector, respectively, and $k_p$ and $k_v$ are the weight of bias for position and velocity, respectively. $\boldsymbol{Q}_t$ is the weight matrix of state cost, and $\boldsymbol{R}$ is the weight matrix of motor cost. The exponential decay term is included because the directional bias need not exist for the entire motion. In our simulation, these parameters were set as follows: $k_p = k_v = 0.5$, $\tau = 130$ (ms). The cost parameters included in $\boldsymbol{Q}_t$ and $\boldsymbol{R}$ were determined to produce trajectories similar to those in the experiments (see S2 Text). The reaching movement was simulated for $0 \leq t \leq T + T_H$ where $T$ is the maximum movement completion time and $T_H$ is the time for which the hand was supposed to hold a position at the target after movement completion (see [23]). $T$ and $T_H$ were set to 400 (ms) and 50 (ms), respectively.

Here, we further extended this idea by introducing a directional bias modulated by trial-by-trial TE (upper panel, Fig 6B). The directional bias is inclined in the opposite direction of TE to reduce it. The direction of the directional bias in the $i$-th trial is represented by $\varphi^i$, the angle from the target direction (clockwise as positive). The TE is equivalent to the directional error represented by $\theta^i$, defined as the angle between the target direction from the start position and the direction from the start position to the endpoint of the reaching. In the presence of TE (i.e., TE > target size), the directional bias is updated according to the directional error as follows:

$$\varphi^{i+1} = b\,\varphi^i - r\theta^i \tag{7}$$

where the constant $b$ is the forgetting rate and is set to 0.95. The constant $r$ is the sensitivity to the degree of the directional bias update to the directional error and set to 0.85. The initial value of the directional bias is 0 (i.e. $\varphi^1 = 0$).

In the absence of TE (i.e., TE < target size), we assumed that the direction bias subtly decays across trials to the original direction towards the target as follows:

$$\varphi^{i+1} = b\,\varphi^i \tag{8}$$

Additionally, we assume that the kinematic plan adaptation is also affected by the motor cost (Eq 6, and see S2 Text) of the generated reaching, and the decay of the directional bias stops, i.e., $b = 1$ when the cost goes below less than 0.01. The threshold value was arbitrarily determined to produce curved null trajectories similar to those in the experiments.

Once a TE greater than the target size occurs, the kinematic bias is active. In contrast, if the TEs keep within the target size throughout the experiment, the kinematic bias remains inactive.

Next, to simulate the internal model adaptation in novel force fields, we changed the value of learning rate $\alpha$. In the adaptation phase, $\alpha$ is increased from 0 to 0.8 such that $\alpha^i = 0.8 \cdot log(log(i)+1)/log(log(155)+1)$ for $1 \leq i \leq 155$. In the de-adaptation phase, $\alpha$ is decreased from 0.8 to 0 in the first 30 de-adaptation trials because de-adaptation process is well known to be much faster than adaptation process[72]. This was given by $\alpha^i = 0.8 \cdot log(log(i-155)+1)/log(log(30)+1)$ for $156 \leq i \leq 185$; $\alpha^i = 0$ for $186 \leq i \leq 305$. The update rule for $\alpha$ was determined to well reproduce the experimental observations.

We simulated the reaching trajectory of the arm modeled as a point mass in the Cartesian coordinates. The movement distance was 150 mm. $B_1$ and $K_1$ were set to 7 Ns/m and 120 N/m for the simulation of VDCF and LIPF, respectively to produce trajectories similar to those in the experiments. PEC was applied over the second half of movement (y > 75 mm) as a one-dimensional spring force (1500 N/m) and damper (100 Ns/m) along x-axis. We discretized the system dynamics with a time step of $\Delta t$ = 10 ms and performed the model simulation in a similar way as that introduced by Izawa et al. [23], except for the directional bias modulated by history of TE. Please see S2 Text for further detail of the model (section of OFC model).

**V-shaped model.** The original model (i.e., flat VS model) assumes that desired trajectory, which the motor system should trace, is a fixed straight line joining the start and target and that the motor command is gradually corrected to reduce the difference between the actual and desired trajectory, which is defined as movement error. In simulation with the model, the error is represented in coordinates of muscle length and written by the form:

$$E = \lambda - \lambda_0 \tag{9}$$

where $E$ is the movement error which is the difference between the actual muscle length, $\lambda$ and the desired muscle length, $\lambda_0$. This error is used to update feedforward command to the individual muscle of the arm on a trial-by-trial basis, based on a simple V-shaped learning function (see S2 Text). The feedforward command for each muscle $k$ is updated from $u_k^i$ to $u_k^{i+1}$ according to the following learning law:

$$
\begin{aligned}
u_k^{i+1}(t) &\equiv [u_k^i(t) + \Delta u_k^i(t + \phi)]_+, \quad [\cdot]_+ \equiv max\{\cdot, 0\} \\
\Delta u_k^i(t) &= \alpha \varepsilon_{k,+}^i(t) + \beta \varepsilon_{k,-}^i(t) - \gamma, \quad [\cdot]_- \equiv [-\cdot]_+ \\
\varepsilon_k^i(t) &= E_k^i(t) + g_d \dot{E}_k^i(t)
\end{aligned}
\tag{10}
$$

where $E_k^i(t)$ is the stretching/shortening in muscle $k$ at time $t$ for trial $i$, and $\Delta u$ is phase advanced by $\phi > 0$, which is feedback delay. $\alpha$ and $\beta$ are the learning parameters ($\alpha > \beta > 0$) and $\gamma$ ($>0$) is a constant de-activation parameter. The term $g_d$ ($>0$) indicates the relative level of velocity error to length error. By implementing this learning law to a 2-joint 6-muscle arm model, Franklin et al. [25] and Tee et al. [73] simulated the reaching trajectories in a broad range of novel force field environments.

Here, we extend the flat model by introducing an idea that the desired trajectory (lower panel in Fig 6B), which is represented in the Cartesian coordinates, is updated according to a trial-by-trial TE in a similar way to the hierarchical OFC model. The desired trajectory is described as a curved line with a deflection, $dx$, 120 mm away from the start position along the y-axis (Fig 6B). Before adaptation, the desired trajectory is the straight line towards the target, that is, $dx$ = 0. In the presence of TE (i.e., TE > target size), $dx$ is updated as follows:

$$dx^{i+1} = bdx^i - rTE^i \tag{11}$$

where the constant $b$ represents the retention of motor learning and is set to 0.95. The constant $r$ to the degree of update of $dx$ to the TE in the previous trial and is set to 0.45. The constant $r$ is the sensitivity to the degree of the desired trajectory update to TE. In the presence of TE, $dx$ is modulated such that the desired trajectory is deflected in the opposite direction to a trial-by-trial TE. The desired trajectory with $dx$ was calculated as the minimum jerk trajectory with the via-point at [dx 120] (mm) from the start position [74].

In the absence of TE (i.e., TE < target size), we assumed that the desired trajectory subtly decays across trials to the original direction towards the target as follows:

$$dx^{i+1} = bdx^i \qquad (12)$$

We again assume that the kinematic plan adaptation is affected by the motor cost of the generated reaching, which is calculated as average muscle tension across all the 6 muscles during movement (see S2 Text). When the cost goes below less than 350, the decay of the desired trajectory stops, i.e., $b = 1$. The threshold value was again arbitrarily determined to produce curved null trajectories similar to those in the experiments.

In simulation, the desired trajectory was converted from the Cartesian to muscle space to apply it to the learning law (Eq 10). The start and target positions were at [0, 350] and [0, 500] (mm) in the Cartesian coordinate (where [0, 0] is at the shoulder joint), respectively. The reach duration was 400 ms. For simplicity, all noise parameters were set to zero. $B_1$ and $K_1$ (see the section of *force fields*) were set to 20 Ns/m and 120 N/m, respectively, to produce trajectories similar to those in the experiments. PEC was applied over the second half of movement ($y > 75$ mm) as a one-dimensional spring force (2500 N/m) and damper (1000 Ns/m) along x-axis. We performed the model simulation in the same way as that introduced by Franklin et al. [25], except that the desired trajectory is modulated by history of endpoint error. Please see S2 Text for further detail of the model (section of V-shaped model).

## Supporting information

**S1 Text. Trajectory adaptation in Experiment-2.**
(DOCX)

**S2 Text. Detail for the simulation.**
(DOCX)

**S1 Fig.** Trajectory adaptation in Experiment-2: (A, C) The hand trajectories of two representative participants and learning curves in PSPF (A) and CPVF (C) averaged across all participants. Note that the scales differ between x and y axes to clearly show trajectory changes along the x-axis. The light gray shades behind some trajectories represent a schematic image of the force field. The adaptation of the TE and LD are shown by traces with open circles and filled circles, respectively. The first 15 TE and LD values are plotted for every single trial, while the subsequent trials (indicated by thick gray lines at the bottom of the figure) are plotted for every five trials. The shaded gray areas around the lines represent standard errors. The light green zones represent the target width (radius: 7.5 mm). (B, D) The TEs and baseline-subtracted LDs in six trial epochs ($1^{st}$, $3^{rd}$-$5^{th}$, $136^{th}$-$155^{th}$ adaptation trials, and $1^{st}$, $3^{rd}$-$5^{th}$, $131^{st}$-$150^{th}$ de-adaptation trials) in PSPF (B) and CPVF (D). Gray dots represent data from individual participants. The error bars indicate standard errors. The light green zone in the TE plots represents the target width.
(TIF)

**S2 Fig.** Simulation results for trajectory adaptation in CPVF, represented by TE (open circle) and LD (filled circle) by the flat (A, B)/hierarchical (C, D) OFC (upper panels) and VS models (lower panels). The flat learning models (only internal model adaptation) were unable to reproduce either the non-monotonic change in LD or the curved null trajectory with a persistent deviation after exposure to CPVF. However, the hierarchical OFC models (kinematic plan learning and internal model adaptation) successfully reproduced both.
(TIF)

## Acknowledgments

We thank Ms. Yuka Furukawa and Ms. Naoko Katagiri for help in recruiting the participants. We thank Dr. Jun Izawa and Dr. Tee for providing the codes used in their studies.

## Author Contributions

**Conceptualization:** Tsuyoshi Ikegami, Gowrishankar Ganesh.

**Data curation:** Tsuyoshi Ikegami.

**Formal analysis:** Tsuyoshi Ikegami.

**Funding acquisition:** Tsuyoshi Ikegami, Gowrishankar Ganesh, Rieko Osu, Mitsuo Kawato.

**Investigation:** Tsuyoshi Ikegami, Gowrishankar Ganesh, Tricia L. Gibo, Toshinori Yoshioka.

**Methodology:** Tsuyoshi Ikegami, Gowrishankar Ganesh, Tricia L. Gibo, Toshinori Yoshioka, Mitsuo Kawato.

**Project administration:** Tsuyoshi Ikegami, Gowrishankar Ganesh.

**Resources:** Tsuyoshi Ikegami, Rieko Osu, Mitsuo Kawato.

**Software:** Tsuyoshi Ikegami, Gowrishankar Ganesh, Toshinori Yoshioka.

**Supervision:** Rieko Osu, Mitsuo Kawato.

**Validation:** Tsuyoshi Ikegami, Gowrishankar Ganesh.

**Visualization:** Tsuyoshi Ikegami.

**Writing – original draft:** Tsuyoshi Ikegami, Gowrishankar Ganesh.

**Writing – review & editing:** Tsuyoshi Ikegami, Gowrishankar Ganesh, Tricia L. Gibo, Toshinori Yoshioka, Rieko Osu, Mitsuo Kawato.

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
