## [Decision Letter · Decision Letter 0]

17 Dec 2020

Dear Dr Ikegami,

Thank you very much for submitting your manuscript "Hierarchical motor adaptations negotiate failures during force field learning" for consideration at PLOS Computational Biology.

As with all papers reviewed by the journal, your manuscript was reviewed by members of the editorial board and by several independent reviewers. The reviewers all agreed that the experiments and model make a potentially valuable contribution to our understanding of motor adaptation. However, they also highlighted a number of limitations of the work and some concerns related to the presentation. In light of the reviews (below this email), we would like to invite the resubmission of a significantly-revised version that takes into account the reviewers' comments.

The reviewers also made a number of suggestions for possible experiments to help address some of the current limitations of the paper. These suggestions would undoubtedly improve the paper if you choose to pursue them. However, it may not be essential to include any additional experiments in your revision if instead you believe the concerns can be adequately addressed by more clearly acknowledging and discussing the limitations of the existing approach.

We cannot make any decision about publication until we have seen the revised manuscript and your response to the reviewers' comments. Your revised manuscript is also likely to be sent to reviewers for further evaluation.

Sincerely,

Adrian M Haith

Associate Editor

PLOS Computational Biology

Samuel Gershman

Deputy Editor

PLOS Computational Biology

Reviewer's Responses to Questions

**Comments to the Authors:**

Reviewer #1: Ikegami and colleagues investigate patterns of adaptation to different types of force fields across several experiments. Adaptation to standard velocity dependent force fields and position dependent force fields are studied. These perturbations differ notably by their magnitude near the end of movement, such that the velocity dependent force field vanishes near the target when velocity decreases, whereas the position dependent force field builds up and increases throughout the reach. As a consequence, there is a difference in end-point error, which is larger for the position-dependent force fields. Different patterns of adaptation and after effects are documented, and it is argued that the terminal target error following position-dependent force fields is a “failure signal”, which plays a key role in adaptation. To account for their results, the authors put forward a computational model that includes changes in the kinematic plan which aims at correcting for the target error, in addition to the adaptation of an internal model to cancel sensory prediction errors.

This is an excellent paper. It provides a comprehensive and rigorous analysis of a robust and reproducible aspect of adaptation to position dependent force fields, and highlights that adaptation strategies may process different error signals during planning and control stages. From a computational perspective, the study is a convincing demonstration that kinematic plans and adaptation can be studied in closed coop control models. This major contribution helps bringing together the often-separated fields of human feedback control and motor adaptation.

In my view, the paper could be published as is. I only have one major concern and a number of minor comments for which I am interested to hear the authors point of view. Some of them a may require clarifications or short discussion points:

Major

My only major criticism is that the movement kinematics varied a lot across experiments and conditions, leading to the possibility that some aspects linked to online control also play a role. For instance, in Experiment 1, the magnitude of the position dependent FF produced larger deviations, and we do not know what would happen if the lateral deviations were comparable across force fields. The same concern holds for the control experiment #3, the virtual spring indeed reduces the target error but it also has an impact on many other kinematic parameters. Of course, it is very difficult if possible to only alter target error while matching all other kinematic parameters, but perhaps a complementary analysis such as subsampling trials with comparable kinematic errors to compare after effect would be useful, otherwise a note of caution seems necessary.

Specific:

Line 64: “motor movement” is a strange formulation.

Intro, paragraph around line 75: The authors imply a strong link between explicit strategies, TE-mediated adaptation, and the fast state of motor adaptation which to my knowledge is not clearly established. One critical aspect is the impact of the paradigm, for instance the explicit strategy that counter target errors following visuomotor rotations to my knowledge does not directly apply to force field learning (perhaps it is a conclusion of this paper). Likewise, it is not clear to me that the implicit adaptation is only linked to the slow state. I would suggest to clarify the argument if I was missing anything, or that these concepts be presented with a bit more caution, keeping in mind that differences across adaptation paradigms (rotations and force fields) my hinder the correspondence between fast-slow, and explicit-implicit components.

Related to the previous point, the possibility of rapid feedback adaptation (Crevecoeur et al., 7(1) ENEURO.0149-19.2019 1–16) is clearly linked to the fast state, but it is a SPE adaptation components in the authors’ terminology and likely linked to implicit adaptation as it produces systematic, one-trial lag after effect in a random scenario.

Lines 91-93: It is a bit of an overstatement that velocity dependent force fields have limited impact in end-point error. Often end-point corrections are not stable and it is not clear which error signal can be used in the case of non-zero terminal velocity or time out errors. Pls consider clarifying that although there is a clear difference in terms of TE magnitude, other aspects of behaviour, such as velocity and ability to stabilise, may also differ.

Lines 108-109: Although it is well defined in the Method section I would recommend giving more info about the force field at this stage (curl field? Orthogonal?)

I was under the impression that the hierarchical nature of the proposed model was not well motivated or not strongly supported in the data. It is fair to assume that a kinematic plan is selected prior to the derivation of the control law, and that the can some sequential organisation of kinematic (TE) and dynamic (SPE) adaptation. But this is potentially a property of the paradigm in which the target is fixed producing apparent hierarchy. However, there is evidence for online adaptation to changes in target location (Braun et al., J Neurosci, 29(20):6472– 6478), thus it is conceivable that such case the update of the kinematic plan be performed downstream of the selection of a control law based on sensory prediction errors. Is the hierarchy truly necessary, or can one imagine that the selection of the kinematic plan and the adaptation of the control law be processed in parallel?

Again on the latter point: the apparent hierarchy is potentially induced by different timescales such that kinematic plan for a fixed target changes from trial to trial whereas adaptation based on sensory prediction errors uses continuous signals. However, in the Crevecoeur et al (cite above eNeuro) and in Braun et al (JNS, cited above), it is suggested that both sensory prediction error and kinematic plans can be updated within movement. Pls consider discussing the potential impact of the time-scales of adaptation to TE and SPE in the hierarchical organisation of the different adaptive components.

Figure 3: the case strongly rests on the fact that long-lasting after effects in terms of lateral deviation characterise washout after position dependent force fields. It seems to be a robust and reproducible feature of the data and clearly the authors’ model can capture this aspect of behaviour, but is it fair to say that it is not clear why it happened in the first place? That it reveals differences between TE and SPE adaptation is fair, but I was under the impression that this finding was not clearly expected a priori.

Please clarify the learning rule or update rule for alpha, was it fitted to the data? Is there a learning algorithm in the sense that the internal model is deduced, or is it just based on partial compensation of the force field with alpha lesser than one?

Pls consider also sharing the simulation code to help the community playing with models of human adaptation and control.

Respectfully submitted,

F. Crevecoeur

Reviewer #2: In this well-written study, the authors ask how motor internal models of environmental dynamics and task error (success/failure) information interact to induce motor learning in human. To this end, they designed and ran a series of well-thought behavioral experiment, which entails participants reaching in various force-fields with their upper limb. While all the force-fields trigger learning at the internal model level, they were cleverly designed to manipulate the amount of task error participants would be exposed to. This enabled the authors to assess the qualitative relationship between internal model and task error. Next, they modelled the observed human behavior with optimal control models to assess the compatibility of the data with distinct working hypotheses: first, that internal model adaptation relies on sensorimotor prediction error alone (“flat” models) or second, that it relies on both sensorimotor prediction error and task error (“hierarchical” models).

While I particularly appreciate the experimental designs and the thoughtful choice of force-fields to tease apart sensory prediction error and task error influence on internal model adaptation, I have two main concerns as detailed below. First, that the behavioral results are presented too much at face value, and that additional controls may benefit the soundness of the results; and second that the modelling section does not really give a “fighting chance” to alternative models, meaning that it does not provide a lot of additional information on top of what the behavioral experiments tell us.

Major points

Lines 78-79: Miyamoto et al (2020) do not claim that the explicit and implicit components of adaptation occur “in parallel” but on the contrary, that they interact. In that sense, that citation would be more fitting several lines down (lines 80-83), where the authors mention work suggesting that task error and sensorimotor prediction error interact. This also applies at lines 475-476.

Behavioral experiments: the LIPF group shows much larger TE errors at the end of the movement than the VDCF group showed LD errors (nearly 4 times more on average based on figure 1). This discrepancy is less pronounced in the PSPF versus CPVF comparison but still present (nearly 2 times more on average from figure S1). This consistent discrepancy may lead to increased feedback gains, or increased stiffness through co-contraction or postural control specifically for the TE groups to help reduce errors. This may explain baseline differences, as least in theory. For instance, a change in postural control may lead to a change in baseline trajectory to accommodate the biomechanics of the arm (Sergio & Scott, Exp Brain Res, 1998). An increase in feedback gains may lead to changes in internal models as hypothesized in Miyamoto, Kawato, Setoyama and Suzuki, (1988, Neural Networks), and shown in humans (Maeda, Cluff, Gribble and Pruszynski, 2018, J Neurosci). The larger errors also suggest participants in these conditions experience stronger forces, which may lead to increased muscle fatigue and explain differences in the new baseline. Ideally the force-fields should be tuned so that the maximal lateral deviation at the distance of interest (LD for VDCF and TE for PIPF) is controlled and matching across conditions. Occasional full channel trials may also help rule out an effect of co-contraction as forces against a channel would signify net torques only. I leave to the authors to decide whether they would include such a control experiment, provide additional analyses to rule out alternative explanations, or explicitly mention this limitation where they deem appropriate in the paper.

Lines 275: “Half (15) of these participants had previously participated in Experiment-1.” Was analysis using this data corrected for multiple comparisons due to being used for several statistical tests? Also, if I understand this right, in line 294 “similar to Experiment-1” is a circular statement since half of the data comes from experiment 1. If the second half shows the same trend, it would be more informative to specify that.

Modelling section: it seems unsurprising and therefore uninformative that a model with two “modules” (a task-error driven module and a sensorimotor-error driven module) can better account for the observed data than a model with only one module, because the data shows two uncorrelated behaviors. Indeed, the authors mention a similar point on lines 712-713. In that sense, the study in its current form supports the incompatibility of the data with a task-error blind (flat) model more than it supports the existence of a hierarchical model of the specific form presented here. Since the behavioral data already make the case for a task-error sensitive model, the current study would greatly benefit from a comparison between different plausible hierarchical models that include task error. The authors already lean in that direction in the discussion on the paragraph at lines 489-503, which was particularly interesting. Indeed, they mention the behavioral data of the current study is incompatible with Kim et al (2019)’s “Adaptation Modulation” model. What about the other two models that Kim and colleagues propose, namely the “Movement Reinforcement” and “Dual Error” models? It may be interesting to simulate each of them and assess their compatibility with the data presented here.

Lines 436 and 439: if the model decays very slowly, then it surely does not “converge” to a new baseline? Could the authors please clarify that point?

Discussion: There is quite a lot of missing work of direct relevance to the current study, although I leave to the authors’ discretion what they deem interesting. The existence of a “new baseline” following reinforcement in visuomotor rotation tasks has been observed many times before (eg Galea et al. 2015 Nat Neuro, Schmuelof et al 2012 J Neurosci), though it remains partially unexplained (Holland et al 2019, J Neurosci), which makes this study potentially exciting. Possible causes that may be relevant are perceptual bias (Vindras et al 1998) and perceptual recalibration (Modchalingam et al 2019), with interesting parallel to the modelling part of this study. An exciting perspective of the current study is that one or both causes are sensitive to task error, which may also be worth pointing out. In line 501-503 the authors mention that the task error in the current study may be explicit, but explicitly reinforced reaching directions can still lead to an implicit new baseline (Holland et al 2019, J Neurosci), which may be worth clarifying. Finally, at lines 402-404, the study from Cashaback et al (2017, Plos Comp Biol) may be of interest to the authors, as it asks this question specifically.

Minor points

Line 83: I find the term “parallel interaction” ambiguous, as “parallel” suggests independent processing, that is, it suggests no interaction. Some rewording may benefit the general reader by removing the ambiguity.

Line 297: “PEC-LIPF” I assume the authors meant “LIPF-PEC”?

Figure S1: I believe the legends (open and closed circle) is missing on this figure? I only find it mentioned in the caption.

Line 610: How long was the hand constrained at the end?

Line 756: Though I understand it is indirectly described in the text, it may be clearer to also write down the cost function for the flat models for the reader’s benefit.

Reviewer #3: In the current manuscript, the authors set out to study how target errors may affect learning in force field adaptation. While force field adaptation has generally been considered the outcome of internal model adaptation, work in visuomotor rotations has suggested that learning can also be the result of explicit re-aiming strategies which appear to be driven by task performance errors (i.e., target errors). Surprisingly, the role of target errors in force field adaptation has received little attention. Here, the authors created force field conditions that would result in target errors. They find that the learning curve under these target-error inducing force field conditions is radically different from standard viscous curl field conditions, which don’t necessarily result in target errors. In addition, they find that movements appear to be curved in the direction of the force field following de-adaptation. A hierarchical model that includes directional changes in the kinematic plan, in addition to force field adaptation, can account for the time course of learning and resulting changes in hand trajectories.

I thought the overarching question of if/how task performance errors alter the learning function is interesting and the findings show a clear behavioral difference, and the simple addition of a directional bias to the modeling simulations is impressive. I have two main comments, one can easily be addressed through the paper’s framing while the other is a bit more difficult as it pertains to the experimental approach.

First, I felt that the narrative and interpretation from the authors could be made clearer. I suspect that the authors were attempting to remain agnostic or cautious, which may have muddled their narrative. I was left wondering whether they think this effect is truly driven by some form of an explicit re-aiming strategy (an interpretative narrative) or if their intent was simply to emphasize an effect of the presence of target errors (a descriptive narrative). This extends to the discussion of their modeling efforts where it is unclear if they think adaptation to the kinematic plan is the automatic result of target errors or reflects volitional changes (i.e., explicit strategy). Again, I suspect this is caution on the authors part, but I think they need to be “explicit” about it so that the reader can clearly know the authors position and, as such, correctly cite the work.

Second, there is a conflation between the physical manipulation and psychological manipulation in the task. We need to be aware that target errors, or task performance errors, are a psychological construction of the participant. Instead of changing the framing of the task, the authors chose to manipulate target errors by the physical properties of the task. In experiment 1, there is a large physical difference between velocity- and position-dependent force field, which would require different control strategies to counter regardless of the psychological evaluation of the participant, yet these are compared as equivalent except for their target error inducing properties. This difference in the physical properties is mitigated by their simulation results, showing that the flat model cannot replicate the learning function of the participants; however, the model when including the directional bias term only qualitatively matches the data. The subtleties that are induced by the physical differences really complicate a convincing, easy-to-see behavioral distinction that could be attributable to a change in task performance criterion/framing.

Their second approach, which is commendable, is to create force field conditions that mix position and velocity perturbations to overcome this issue but here the data here are not convincing either. There’s a numerical (if not significant) bias in de-adaptation in the position-skewed-position dependent field. This could be a spurious finding or it could be power issue. I would also suggest that these experiments be fully presented in the manuscript rather than only partially (with the majority in the supplemental). Their final approach is to use a partial error clamp over the second half of reach to remove target errors. Again, though this requires a change in the physical aspects of the task and, thus, it is conflated with psychological interpretation of target error.

I would suggest that the authors take an approach where they simply change the participants’ psychological/conceptual framing of task performance instead of through the perturbations. This could easily be accomplished through instruction, changing the target size, providing points or monetary incentive, task performance criterion etc. This would also help clarify if the directional bias they observe is the result of a control strategies in response to position-dependent field induced target error or if the directional bias arises simply from a volitional change in aiming direction by the subject. In the end, I appreciate the intention of the current study and found the model to be surprisingly useful, but I think a more psychological approach to manipulating task performance could overcome confounds with changing the perturbations.

**Have all data underlying the figures and results presented in the manuscript been provided?**

Reviewer #1: Yes

Reviewer #2: Yes

Reviewer #3: Yes

PLOS authors have the option to publish the peer review history of their article (what does this mean?). If published, this will include your full peer review and any attached files.

Reviewer #1: **Yes: **Frederic Crevecoeur

Reviewer #2: No

Reviewer #3: No
---

## [Decision Letter · Decision Letter 1]

24 Mar 2021

Dear Dr Ikegami,

We are pleased to inform you that your manuscript 'Hierarchical motor adaptations negotiate failures during force field learning' has been provisionally accepted for publication in PLOS Computational Biology.

Regarding Reviewer 3's remaining concerns: I agree that there are a number of outstanding questions with regard to how the target errors induced by the force field approach relate to other manipulations that have been used in the past. A more systematic exploration of this question would certainly be valuable going forwards. However, I feel the issue has been adequately discussed in the paper and further experiments are not necessary at this stage.

Best regards,

Adrian M Haith

Associate Editor

PLOS Computational Biology

Samuel Gershman

Deputy Editor

PLOS Computational Biology

Reviewer's Responses to Questions

**Comments to the Authors:**

Reviewer #1: I am satisfied with the revised version of the paper and recommend publication of the manuscript.

Reviewer #2: Response to the authors

After careful reading of the responses the authors provided to my original comments, I find the amendments they made satisfactory in either addressing them directly or mentioning their content explicitly in the final work. Consequently I have no further points to raise and am happy to support it for publication.

Reviewer #3: While I thought that the flow of the manuscript was much improved, I still think the authors have a fundamental problem/conflation in the study design that complicates interpretation. Target errors are inextricably linked to the dynamics of the perturbation. Thus, it is impossible to determine if the pattern of results, especially the changes to the null-field trajectory, are the result of a task-performance sensitive process (i.e., strategy) or a lower-level process concerned with kinematic planning, which may be distinguishable from a strategy. As I said in my previous review, I think this issue could easily be addressed with a follow-up experiment that dissociates target errors from the force perturbation, such as changing the size of the target or jumping the target. Findings from a study like this could help constrain the modeling efforts and scope of speculation in the manuscript.

**Have all data underlying the figures and results presented in the manuscript been provided?**

Reviewer #1: Yes

Reviewer #2: Yes

Reviewer #3: Yes

PLOS authors have the option to publish the peer review history of their article (what does this mean?). If published, this will include your full peer review and any attached files.

Reviewer #1: No

Reviewer #2: No

Reviewer #3: No

---

## [Editor Report · Acceptance letter]

12 Apr 2021

PCOMPBIOL-D-20-02060R1 

Hierarchical motor adaptations negotiate failures during force field learning

Dear Dr Ikegami,

I am pleased to inform you that your manuscript has been formally accepted for publication in PLOS Computational Biology. Your manuscript is now with our production department and you will be notified of the publication date in due course.

With kind regards,

Katalin Szabo
